# Alternative splicing of apoptosis genes promotes human T cell survival

**Davia Blake**[1,2]**, Caleb M Radens**[2]**, Max B Ferretti**[2]**, Matthew R Gazzara**[2,3]**, Kristen W Lynch**[1,2]*****

[1]Immunology Graduate Group, University of Pennsylvania, Philadelphia, United States; [2]Department of Biochemistry and Biophysics, University of Pennsylvania, Philadelphia, United States; [3]Department of Genetics, University of Pennsylvania, Phildelphia, United States

**Abstract** Alternative splicing occurs in the vast majority of human genes, giving rise to distinct mRNA and protein isoforms. We, and others, have previously identified hundreds of genes that change their isoform expression upon T cell activation via alternative splicing; however, how these changes link activation input with functional output remains largely unknown. Here, we investigate how costimulation of T cells through the CD28 receptor impacts alternative splicing in T cells activated through the T cell receptor (TCR, CD3) and find that while CD28 signaling alone has minimal impact on splicing, it enhances the extent of change for up to 20% of TCR-induced alternative splicing events. Interestingly, a set of CD28-enhanced splicing events occur within genes encoding key components of the apoptotic signaling pathway; namely caspase-9, Bax, and Bim. Using both CRISPR-edited cells and antisense oligos to force expression of specific isoforms, we show for all three of these genes that the isoform induced by CD3/CD28 costimulation promotes resistance to apoptosis, and that changes in all three genes together function combinatorially to further promote cell viability. Finally, we show that the JNK signaling pathway, induced downstream of CD3/CD28 costimulation, is required for each of these splicing events, further highlighting their co-regulation. Together, these findings demonstrate that alternative splicing is a key mechanism by which costimulation of CD28 promotes viability of activated T cells.

**\*For correspondence:**
klync@pennmedicine.upenn.edu

## Editor's evaluation

Blake and colleagues examine programs of alternative splicing in apoptotic proteins controlled during T-cell activation. Apoptotic regulators have long been known to often be expressed in pairs of pro- and anti-apoptotic isoforms. The demonstration of how a program of these splicing changes contributes to immune responses is significant to the understanding of both apoptosis and T-cell biology.

## Introduction

Alternative splicing is a co-transcriptional process that consists of the regulated inclusion of exonic or intronic regions of the pre-mRNA transcript during RNA processing. In human cells, almost all multi-exon genes express multiple mRNA isoforms as a result of alternative splicing and these isoforms often exhibit differential degradation or translation rates, localization patterns, or even encode for proteins with differing activity (*Baralle and Giudice, 2017*; *Braunschweig et al., 2013*; *Ule and Blencowe, 2019*). Consequently, differences in the ratio of isoforms generated by alternative splicing can impact overall cellular activity (*Baralle and Giudice, 2017*; *Ule and Blencowe, 2019*).

Some of the best characterized examples of how alternative splicing can switch protein activity and cellular function come from the apoptosis signaling pathway (*Blake and Lynch, 2021*). Apoptosis is an immunologically silent form of cell death in which cells undergo shrinkage, phosphatidylserine exposure on the extracellular membrane, DNA fragmentation, and cellular blebbing (*Elmore, 2007*). Triggering of apoptosis by cell-extrinsic factors is initiated by the activation of death receptors on the cell surface, such as Fas (CD95) (*Guicciardi and Gores, 2009*; *Strasser et al., 2000*). Notably, skipping of exon 6 of the gene encoding FAS is a commonly observed pattern of alternative splicing (*Cheng et al., 1994*; *Izquierdo et al., 2005*). This skipping of exon 6 removes the transmembrane domain and results in a soluble form of FAS that is secreted to the extracellular environment and functions as a dominant negative competitor to block FAS ligand-induced apoptosis (*Cheng et al., 1994*).

Alternative splicing also regulates the activity of members of the two major protein families, Caspases and BCL2 proteins, that mediate downstream signaling pathways downstream of FAS leading to apoptosis (*Czabotar et al., 2014*; *Jin and El-Deiry, 2005*; *McIlwain et al., 2013*; *Strasser et al., 2000*). Caspases are cysteine proteases, while BCL2 family members are made up of pro- and anti-apoptotic players that work in balance to control mitochondrial membrane permeabilization (*Czabotar et al., 2014*; *Singh et al., 2019*). Upon oligomerization of death receptors, pro-caspase 8 is recruited to the intracellular domain of the death receptor, which promotes its dimerization and cleavage to the active Casp8 form (*Guicciardi and Gores, 2009*). Activated Casp8 then cleaves and activates downstream effector caspases to induce cell death, as well as cleaving the BCL2 family member, Bid, to produce the cleavage product tBid (*Singh et al., 2019*). tBid, in turn, promotes intrinsic signaling events that culminate in the breakdown of the mitochondrial outer membrane permeabilization (MOMP) and the subsequent release of cytochrome *c* through the pore formation of effector BCL2 family proteins, Bax and Bak (*Billen et al., 2008*; *Ly et al., 2003*). Some BCL2 family proteins like Bim (also called BCL2L11) can promote the Bax and Bak activity, while others, like an isoform of Bcl-x, can inhibit MOMP events. Like FAS, the gene encoding Bcl-x is alternatively spliced to produce two isoforms: Bcl-xS, which promotes the activity of Bax and apoptosis, and a longer isoform, Bcl-xL that inhibits Bax activity and thus is anti-apoptotic (*Billen et al., 2008*; *Stevens and Oltean, 2019*). Bax and Bim also have reported alternatively spliced isoforms; however, the functional relevance of these is less well characterized (*Blake and Lynch, 2021*). Finally, once released from the mitochondria, cytosolic cytochrome *c* activates the enzymatic activity of the initiator caspase-9, through formation of the apoptosome, which then also cleaves the aforementioned effector caspases to promote apoptosis (*McIlwain et al., 2013*; *Singh et al., 2019*). Caspase-9 activity is also regulated by alternative splicing, as skipping of the exons encoding the large catalytic domain of caspase-9 result in a dominant negative form of the protein that inhibits apoptosis (*Seol and Billiar, 1999*). Therefore, regulating the isoform expression of apoptotic molecules such as FAS, Bcl-x, and caspase-9 is a critical mechanism cells use to control their sensitivity to apoptotic stimuli.

Regulation of apoptosis is particularly critical for the development and function of human T cells (*Krammer et al., 2007*; *Zhan et al., 2017*). During negative selection in the thymus, self-reactive thymocytes undergo apoptosis to promote tolerance to self (*Penninger and Mak, 1994*). Induction of apoptosis also limits the inflammatory activity of T cells during the contraction phase of immune perturbance, through the mechanism of activation-induced cell death (*Krammer et al., 2007*; *Zhan et al., 2017*). Finally, sub-optimal induction of the T cells often arrests cells in a state known as anergy, in which cells become unresponsive to further stimuli and subsequently undergo apoptosis (*Fathman and Lineberry, 2007*; *Lechner et al., 2001*). Anergy is known to be induced through the stimulation of the T cell receptor (TCR) in the absence of costimulation of the CD28 receptor. By contrast, when the TCR is normally activated by recognizing foreign peptides presented by an antigen presenting cell (APC), the APC also induces costimulation of CD28, which acts together with TCR engagement to activate a signal transduction cascade to stimulate the T cell (*Fathman and Lineberry, 2007*; *Watts, 2010*). CD28 costimulation enhances the expression of many genes in activated T cells to a level greater than that achieved via engagement of the TCR alone (*Diehn et al., 2002*; *Riley et al., 2002*; *Watts, 2010*).

Genes regulated by CD28 costimulation include the pro-proliferative and pro-survival cytokine IL2, and the anti-apoptotic isoform Bcl-xL described above. Current models hold that increased expression of IL2 and Bcl-xL is a primary mechanism by which CD28 costimulation prevents anergy and induces survival and proliferation of effector T cells (*Diehn et al., 2002*; *Riley et al., 2002*; *Watts,*

2010). However, the potential contribution of CD28 costimulation to regulate cell survival through alternative splicing has not been explored. More generally, while it is well established that stimulation of T cells leads to extensive changes in alternative splicing, the only study thus far to investigate the specific contribution of CD28 signaling to alternative splicing used microarrays, an older technology not optimized for quantifying splicing, and results were not confirmed by orthogonal assays (*Butte et al., 2012*). One apoptosis-related gene shown to conclusively undergo splicing regulation upon T cell activation is FAS (*Izquierdo et al., 2005*); however, it is unknown if this splicing event is impacted by CD28 costimulation. Therefore, the degree to which CD28 costimulation generally regulates alternative splicing, and the functional contribution of alternative splicing broadly to the control of apoptosis in T cells, both remain largely unknown.

Here, we investigate the role of CD28 costimulation by high-depth RNA-sequencing in primary CD4+ T cells, stimulated through the TCR with or without CD28. We find that while TCR stimulation alone is sufficient to induce changes in splicing, CD28 costimulation greatly amplifies the extent of this change for 10–20% of splicing events, analogous to the impact of CD28 signaling on gene expression. We further show that CD28 signaling enhances alternative splicing of several Bcl-2 family members and caspases that are critical regulators of apoptosis, and we demonstrate that the splicing events enhanced by CD28 costimulation increase cellular resistance to apoptotic stimuli. Finally, we show that costimulation coordinately regulates the splicing of caspase-9, Bax, and Bim through the JNK signaling pathway and these alternative splicing of these genes cooperatively enhance cell survival. Therefore, we conclude that CD28 costimulation promotes cell survival not only by enhancing the transcriptional program induced by TCR signaling, but also by promoting the expression of anti-apoptotic isoforms via regulated alternative splicing.

## Results

### Temporal regulation of alternative splicing in primary human CD4+ T cells controls a distinct set of genes than are regulated by expression

To better understand the breadth and dynamics of alternative splicing changes following T cell activation, we sought to characterize global alternative splicing changes induced at early and late time-points following T cell stimulation, as well as to determine the impact of CD28 costimulation on splicing (*Figure 1A*). We first isolated primary human CD4+ CD45RO- T cells from three healthy human donors, varying in age and gender, with >90% (CD4+) and >85%(CD45RO) purity (*Figure 1—figure supplement 1A-C*). These cells were then stimulated in culture with either anti-CD28, anti-CD3, or anti-CD3/CD28 antibodies for either 8 or 48 hr to analyze early and late alternative splicing changes. Following stimulation, total RNA was isolated from cells and subjected to poly(A) selected RNA-sequencing to a depth of 65 million read per sample, which allows for rigorous quantification of splicing (*Figure 1A*). Potential variability of T cell activation between human donors was controlled for by the expression of the activation marker, CD69, which is routinely used to measure the robustness of T cell activation, and IL2 expression, which is well characterized to be enhanced by the presence of CD28 costimulation (*Diehn et al., 2002*; *Riley et al., 2002*; *Watts, 2010*). Importantly, by flow cytometry, CD69 was consistently upregulated between donors when T cells were stimulated by either CD3 or CD3/CD28 for 48 hr (*Figure 1B*). In addition, IL2 expression was enhanced to a similar extent in all donors in the context of CD28 costimulation when compared to CD3 stimulation alone (*Figure 1C*). We also carried out differential gene analysis to quantify and confirm that the number of genes that are differentially expressed in our dataset is comparable to the numerous other datasets recently published. Indeed, we are able to detect over 1600 and 2100 genes that are differentially expressed by CD3 and CD3/CD28 conditions at 48 hr of T cell stimulation, respectively (*Figure 1—figure supplement 1D-E*), consistent with other studies demonstrating that CD28 signaling enhances the expression of a subset of genes (*Diehn et al., 2002*; *Riley et al., 2002*; *Watts, 2010*). In addition, we observe clear upregulation of many genes that are described to have increased expression upon T cell activation, like interferon gamma and TNF (*Figure 1—figure supplement 1D*, *Supplementary file 1*).

To quantify splicing differences between conditions, we used the MAJIQ algorithm, which is optimized for the quantification of both previously annotated and novel alternative splicing events, including those that are simple/binary splicing changes or complex splicing events (*Vaquero-Garcia*

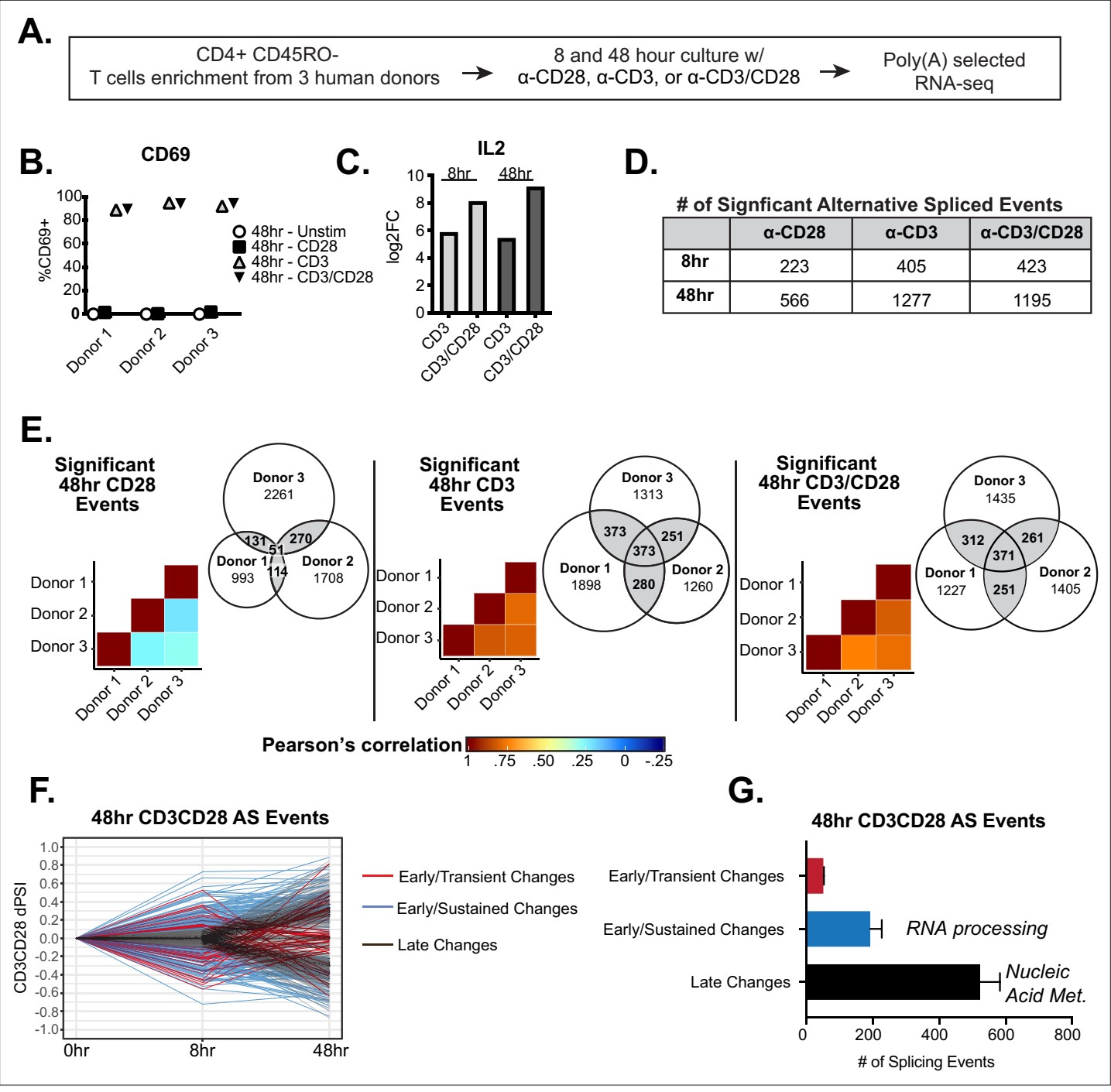

**Figure 1.** Temporal regulation of alternative splicing in primary human CD4+ T cells controls a distinct set of genes than are regulated by expression. (**A**) Schematic of experimental set-up for RNA-sequencing (RNA-seq). (**B**) Flow cytometry analysis of CD69 cell surface expression on stimulated primary CD4+ CD45RO- T cells cultured for 48 hr. (**C**) Log2 fold changes (stimulated/unstimulated) of IL2 expression quantified by RNA-seq analysis. (**D**) Number of significant (dPSI >10%, p<0.05) splicing changes induced by either anti-CD28, anti-CD3, or anti-CD3CD28 antibodies for 8 or 48 hr of culture. dPSI is equivalent to stimulated PSI – unstimulated PSI for the corresponding timepoint (see Materials and methods). (**E**) Pearson's correlation analysis of significant splicing events in the specified stimulation conditions compared between human donors. Venn diagrams display the overlap of significant splicing events induced by the specified stimulation condition compared between human donors. (**F**) Splicing events significantly (dPSI >10%, p<0.05) by the 48 hr-CD3CD28 condition were analyzed for differential temporal changes between 8 and 48 hr of culture. Events labeled as 'Early/Transient Changes' have a significant change at 8 hr but a decrease in dPSI of >2 by 48 hr, while 'Early/Sustained Changes' were significant at 8 hr and showed no decrease of more than twofold by 48 hr. Lastly, remaining events were labeled as 'Late Changes'. (**G**) Averages of the number of significant events

*Figure 1 continued on next page*

*Figure 1 continued*

categorized based on their temporal patterns as seen in **F**, between human donors. Bar graph depicts mean +/-SEM for the 3 donors. Summary of top gene ontology (GO) categories presented by splicing events within 'Early/Sustained Changes' and 'Late Changes'.

The online version of this article includes the following figure supplement(s) for figure 1:

**Figure supplement 1.** Additional information on initial RNA-sequencing (RNA-seq) analysis.

**Figure supplement 2.** Change in splicing as binned by initial percent spliced isoform (PSI).

**Figure supplement 3.** Differential gene expression induced by CD28 costimulation.

*et al., 2016*). In the MAJIQ analysis the abundance of any given splicing pattern (isoform) relative to the total is quantified as percent spliced isoform (PSI), with changes in isoform abundance between conditions quantified as a deltaPSI or dPSI (*Vaquero-Garcia et al., 2016*; *Supplementary file 2*). At 8 hr of stimulation, MAJIQ identifies 223, 405, and 423 significant alternative splicing changes (>10% dPSI, >95% probability in at least two of three donors) in the CD28, CD3, and CD3/CD28 stimulation conditions, respectively. Upon 48 hr of stimulation, the number of significant alternative splicing changes increased to 566, 1277, and 1195 alternative splicing changes in the CD28, CD3, and CD3/CD28 stimulation conditions, respectively (*Figure 1D*). The usage of a 10% dPSI cut-off to define significant splicing changes is based on previous studies demonstrating this cut-off to identify splicing changes that are highly validated by orthogonal methods and also suggests a meaningful biological phenotype (*Vaquero-Garcia et al., 2016*). We also excluded the possibility that our identified stimulation-induced splicing events were dependent on skewed PSI distributions that exist prior to stimulation as we observe no strong biases of the distribution of unstimulated PSI values for those splicing events that exhibit stimulation-induced changes (*Figure 1—figure supplement 2*). Importantly, we observe an overall high correlation (Pearson >0.5) in the splicing induced by either CD3 or CD3/CD28 across in all donors (*Figure 1E*). By contrast, the splicing changes observed upon stimulation with CD28- alone are much more variable (*Figure 1E*, left panel). To ensure focus on the most robust and reproducible changes, our subsequent studies are restricted to splicing events induced by CD3 or CD3/CD28 that met the threshold criteria (>10% dPSI, >95% probability) in at least two of three donors (*Figure 1D and E*) while we have disregarded events induced by CD28 alone for the rest of the analysis due to overall low correlation rates between donors.

In all cases, the predominant splicing pattern observed upon T cell stimulation was differential inclusion of a cassette exons, as has typically been observed in other studies of signal-induced splicing regulation (*Ajith et al., 2016*; *Thompson et al., 2020*; *Figure 1—figure supplement 1F, G*). Other common patterns of splicing were also observed, such as differential use of alternative 5' and 3' splice sites. Finally, approximately 30% of all splicing changes observed were non-binary/complex events, consistent with other studies using MAJIQ (*Vaquero-Garcia et al., 2016*). Moreover, in line with other global transcriptome studies (*Agosto et al., 2019*; *Shinde et al., 2017*; *Thompson et al., 2020*), we detect only a minimal overlap between genes regulated by at the level of expression and those regulated by alternative splicing (*Figure 1—figure supplement 1I*). Therefore, alternative splicing regulates a unique set of genes upon T cell activation, which have the potential to regulate T cell function independently of effects on transcript abundance.

Having established a reliable set of splicing changes at individual conditions and timepoints, we next sought to investigate changes in splicing regulation over time by selecting those splicing events that are significant at 48 hr of CD3/CD28 T cell stimulation and plotting the degree of splicing change (dPSI) at 48 hr against the detected amount of change at 8 hr of stimulation (*Figure 1F*, *Figure 1—figure supplement 1H*). We then categorized the differential trends seen between timepoints, namely: events that are upregulated by 8 hr and return to baseline or beyond by 48 hr ('Early/Transient Changes'), events that are upregulated by 8 hr and sustained at 48 hr ('Early/Sustained Changes'), and events that are not changing until 48 hr of stimulation ('Late Changes') (*Figure 1F–G*, *Supplementary file 3*). Interestingly, most events display 'Late Changes' patterns and are enriched for gene ontology (GO) categories that involve RNA metabolic processes (*Figure 1—figure supplement 1J*). The next biggest subset are events that are 'Early/Sustained Changes' are enriched for GO terms that mostly involve RNA splicing and processing categories (*Figure 1—figure supplement 1K*). Alternative splicing events that are 'Early/Transient' are not enriched for any specific GO categories. Therefore, splicing events that exhibit different patterns of behavior between 8 and 48 hr of T cell

activation are clearly distinct and may regulate different aspects of T cell biology. We note that overall gene expression follows a similar pattern, in that over 80% of genes that change expression at 48 hr do not show notable increases in expression at earlier (8 hr) timepoints. Interestingly, we identify over 300 genes encoding RNA binding proteins (RBPs) that exhibit differential expression following costimulation, including 60 genes that change rapidly upon activation (*Figure 1—figure supplement 3*; *Supplementary file 1*). It is likely that some, if not many, of these changing RBPs drive the changes in splicing we observe; however, analysis of sequences surrounding the regulated splicing events failed to identify any enriched motifs corresponding to binding sites for the changing RBPs (*Supplementary file 4*). The lack of clear motif enrichment likely reflects a complex network of regulation as discussed below (see Discussion).

## CD28 enhances a subset of alternative splicing events upon primary human CD4+ T cell activation

Beyond temporal differences, we are also interested in the impact of CD28 costimulation on splicing. As discussed above, despite the clear impact of CD28 on transcription, the role of CD28 costimulation on splicing regulation in primary human CD4+ T cells remains largely unstudied. The number of genes in which we observe alternative splicing upon T cell activation is roughly similar with or without CD28 costimulation (*Figure 1D*). Moreover, we find that CD3/CD28- and CD3-induced splicing changes are generally correlative at 8 hr of stimulation ($R^2$ mostly >0.5), with even greater correlation at 48 hr ($R^2 > 0.75$) (*Figure 2A–D*). These results suggest that there is not a broad requirement for costimulation in splicing regulation during CD3 engagement of T cells. However, to determine if CD28 costimulation might have an impact on a subset of splicing events, we calculated the ratio between the extent of splicing changed induced by CD3/CD28 vs. CD3 alone (CD3/CD28 dPSI/CD3 dPSI) to analyze the effects of costimulation at the individual splicing event level. To prevent CD3-induced splicing events with a very low dPSI (<1%) from skewing the ratio quantification, we normalized all CD3-induced dPSI values of <1% to 1%. Interestingly, at 8 hr, 24% of alternative splicing events have a CD3CD28/CD3 dPSI ratio greater than 2, suggesting that the amount to which these splicing events are regulated is markedly enhanced by CD28 costimulation (*Figure 2E*). By 48 hr we observe a smaller effect of costimulation, with only ~7% of alternative splicing events exhibiting a CD3CD28/CD3 dPSI ratio greater than 2 (*Figure 2F*). We note that alternative splicing events regulated by CD28 costimulation are largely those for which CD3 induces a change in splicing between 1% and 25% dPSI, therefore CD28 signaling produces a robust splicing change primarily for events that exhibit minimal changes with CD3 alone (*Figure 2—figure supplement 1*). We also note that most of the splicing events regulated by costimulation at 8 hr are subsequently downregulated by 48 hr of stimulation (i.e. 'Early/Transient'), while most of the costimulation-enhanced events observed at 48 hr are 'Late Changing' events and not regulated at 8 hr of stimulation (*Figure 2—figure supplement 1*). Therefore, the spectrum of splicing events regulated by CD28 costimulation at 8 and 48 hr following T cell stimulation are distinct from one another, rather than CD28 costimulation simply 'speeding up' the change in splicing.

Given that the set of genes we are analyzing for changes in splicing are those for which no change in expression is observed, we can rule out a model in which CD28 signaling impacts splicing as a secondary effect of changes in transcription or abundance of a given gene. To begin to understand how CD28 costimulation might directly impact splicing, we asked if the expression of any RBPs is induced by CD28. While we don't observe any RBPs whose expression is clearly impacted by CD28 at 8 hr, we identify 49 RBP-encoding genes that are expressed at least twice as highly upon CD28 costimulation than is observed downstream of CD3 engagement alone (*Figure 1—figure supplement 3C, D*). Notably, for at least two of these RBPs with known splicing regulatory activity, SRSF1 and hnRNPA2B1, we observe enrichment of their cognate binding motif around CD28-enhanced alternative splicing events (*Figure 2G*; *Supplementary file 4*). We acknowledge that more RBPs may be regulated by CD28 costimulation through post-transcriptional or post-translational mechanisms. Therefore, further analysis is necessary to determine if unaccounted RBPs could also bind to enriched motif sequences and confer CD28-induced splicing changes. Nevertheless, these data demonstrate that CD28 signaling impacts gene expression in activated T cells not only through induced expression but also via direct impact of alternative splicing, and this latter effect is likely driven by increased activity of a subset of splicing regulators, including SRSF1 and hnRNPA2B1.

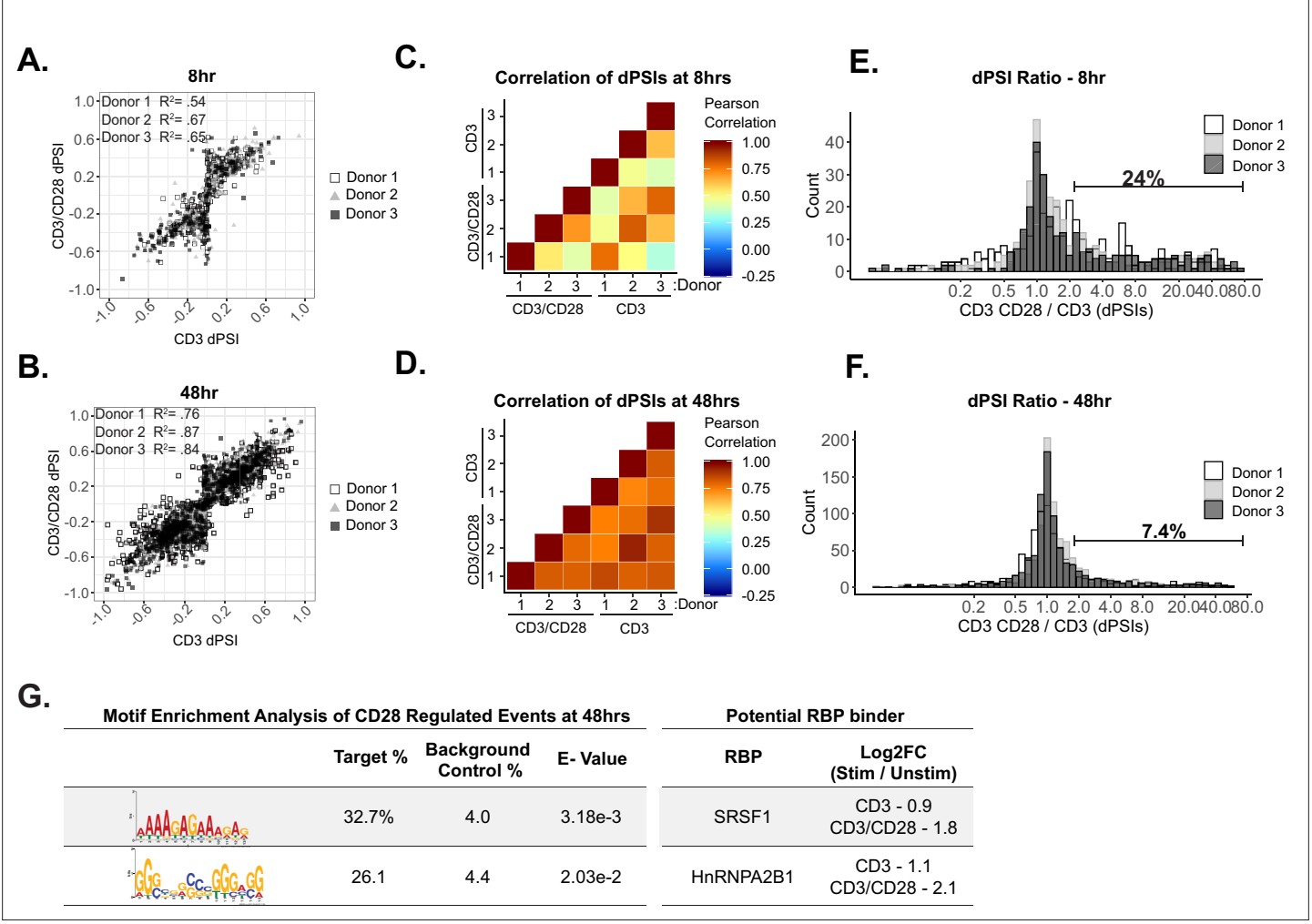

**Figure 2.** CD28 enhances a subset of alternative splicing events upon primary human CD4+ T cell activation. (**A–B**) Dot plots displaying correlation trends of splicing changes induced by anti-CD3 and anti-CD3CD28 antibodies in stimulated human primary CD4+ CD45RO- T cells in their respective timepoints. (**C–D**) Pearson's correlation analysis of splicing quantifications between induced by anti-CD3 and anti-CD3CD28 stimulation of human primary CD4+ T cells in their respective timepoints. (**E–F**) Histograms depict the ratio of anti-CD3CD28 percent spliced isoform (PSI) over anti-CD3 PSI to quantify the number of splicing changes enhanced by CD28 costimulation. Splicing events with a ratio >2 are defined as enhanced by CD28 costimulation. (**G**) (*Left*) Motif enrichment analysis of sequences that are enriched within splicing events regulated by CD28 costimulation at 48 hr of culture, as seen in **F**. (*Right*) Examples of RNA binding proteins that are regulated by CD28 costimulation based on RNA-seq expression analysis (***Supplementary file 1***) and have a consensus binding sequences similar to enriched sequence motifs.

The online version of this article includes the following figure supplement(s) for figure 2:

**Figure supplement 1.** Further information on CD28 costimulation-induced events.

## Splicing events enhanced by CD28 costimulation include those in genes involved in controlling apoptosis

In the case of induced transcription, CD28 costimulation increases the expression of several genes involved in T cell proliferation and effector function, thus providing a mechanistic basis for at least some of the requirement for costimulation to achieve a full T cell immune response. Importantly, we find that genes that exhibit CD28-induced alternative splicing in the analysis above also include genes with critical roles in T cell function. For example, at 8 hr CD28 costimulation increases the production of the full-length isoforms of BPTF, STRADA, and TAB3 (***Figure 3A-C***), all proteins that have the potential for influencing maintenance of T cell numbers and function (***Wu et al., 2016***; ***Yin et al., 2012***) or NF-kB activity (***Jin et al., 2004***). Similarly, splicing events in genes encoding caspase-9 (caspase-9), IL12RB, and MR1 displayed sensitivity to CD28 costimulation by 48 hr of T cell stimulation

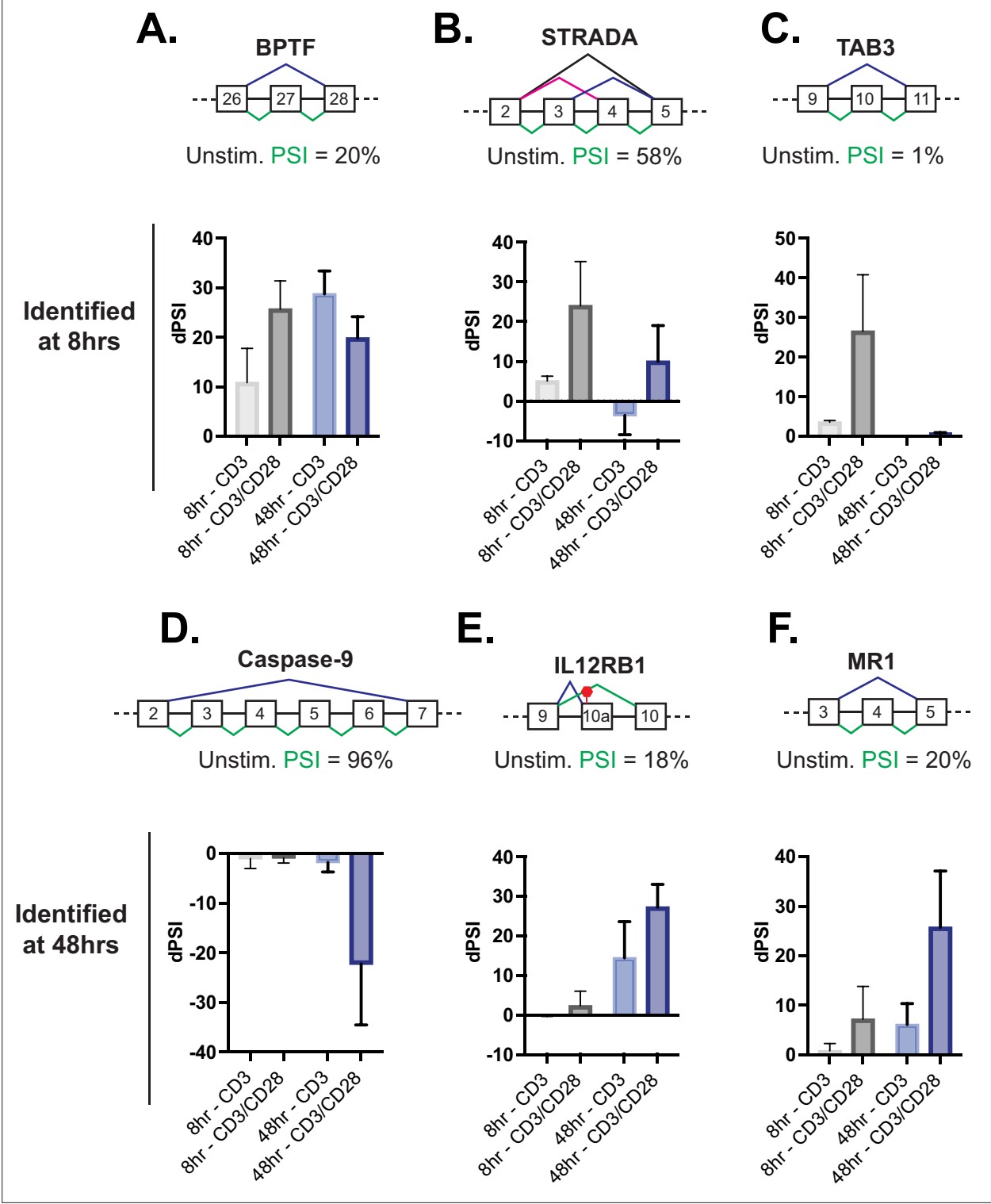

**Figure 3.** Splicing events enhanced by CD28 costimulation include those in genes involved in controlling apoptosis. (**A–C**) Examples of genes regulated by CD28 costimulation at 8 hr of T cell stimulation, as quantified in *Figure 2E*. Each schematic displays the baseline percent spliced isoform (PSI) level from unstimulated cells and the splicing pattern quantified as PSI in green. The average deltaPSI (dPSI) levels in anti-CD3 and anti-CD3/CD28 stimulation conditions across three human donors, as determined by MAJIQ analysis of RNA-seq are in the graph below. Error bars represent SEM across the 3 donors. (**D–F**) Same as **A-C**, except showing examples of genes regulated by CD28 costimulation at 48 hr of T cell stimulation, as quantified in *Figure 2F*.

(*Figure 3D–F*). Loss of IL12RB expression following T cell activation biases differentiation toward the Th1 fate (*Kano et al., 2008*), while major histocompatibility complex class I-related (MR1) is recognized by and activates a unique type of T cells with an invariant TCR alpha chain (*Flores-Villanueva et al., 2020*). However, given the role of CD28 costimulation in promoting T cell viability, we were most intrigued by the influence of CD28 on the alternative splicing of caspase-9 (*Figure 3D*).

The splicing event within caspase-9 induced by CD28 costimulation is of particular interest because of the potential of the splicing event to control T cell survival (*Figure 4A*). Specifically, the RNA-sequencing analysis reveals a large and CD28-dependent increase in the skipping of caspase-9 exons 3–6 upon T cell activation at 48 hr (*Figure 3D*), which we further validated by RT-PCR analysis of RNA harvested from addition donors (*Figure 4B*). As discussed above, exons 3–6 of the caspase-9 gene encode a portion of the catalytic domain responsible for the subsequent cleavage of caspase-3 to effect apoptosis (*Figure 4A*). Skipping of these exons leads to production of a dominant negative form of the protein that, at least in lung carcinoma cell lines, has been shown to protect against apoptosis (*Goehe et al., 2010*; *Seol and Billiar, 1999*; *Shultz et al., 2010*).

Given the well-described functional impact of caspase-9 splicing changes, and the potential implication for T cell activity, we investigated if other genes within death signaling pathways are also regulated by alternative splicing upon T cell activation as well as their sensitivity to CD28 costimulation. Notably, out of 14 additional splicing events in known apoptotic genes that were quantified by MAJIQ in our data, the genes Bim, Bax, and caspase-1 displayed the largest splicing changes (>20% dPSI) upon 48 hr of primary CD4+ T cell activation (*Figure 4—figure supplement 1*), which we confirmed by RT-PCR (*Figure 4C–E*). Moreover, in each of these three cases, as for caspase-9, we observe a decreased expression of the longer isoform upon T cell activation and this change is at least somewhat enhanced by CD28 costimulation (*Figure 4C–E*). The splicing event within Bim includes three isoforms, Bim(EL), Bim(L), and Bim(S), that are discriminated by the variable inclusion of intron 2b and exon 3c, which encode the N-terminal region of the protein (*Figure 4C*). Bax splicing involves the decreased inclusion of exon 3 in activated T cells, which encodes the BH3 domain of the protein, necessary for its oligomerization and activation (*Figure 4D*). Lastly, the splicing event within caspase-1 identified in our MAJIQ analysis represents a novel intron retention event in the 3′UTR region of the transcript (*Figure 4E*). This change in the 3′UTR correlates with reduced caspase-1 protein expression, suggesting the potential for translation regulation (*Figure 4F and G*). Critically, caspase-9, Bim, Bax, and caspase-1 are not significantly regulated at the level of gene expression upon T cell activation, although other apoptotic-related genes such as FasLG, Fas, and Bcl-x do exhibit increased expression at 48 hr after stimulation (*Figure 4—figure supplement 1*), as described previously (*Boise et al., 1995*; *Brunner et al., 1995*). However, the splicing changes in at least caspase-9 and Bim do result in an increase in the abundance of the smaller protein isoforms while the contribution of splicing changes to regulate protein isoforms of Bax and caspase-1 remain unclear. The differences in the extent of detected protein changes may reflect differences in the kinetics of splicing of caspase-9, Bim, Bax, and caspase-1, in which Bax is the most delayed response following T cell activation (*Figure 4—figure supplement 1*).

## Activation-induced alternative splicing of apoptotic genes promotes T cell survival

Similar to caspase-9, Bim and Bax are well documented to regulate intrinsic signaling events within the apoptosis signaling pathway (*Figure 4A*), thus we hypothesized that the splicing of these events all might control cell survival following T cell activation. To test this hypothesis and better understand the phenotypic impact of splicing events in T cells, we used CRISPR/Cas9 to force an increase in the activation-induced (i.e. smaller) isoform of caspase-9, Bim, or Bax within Jurkat T cells, by deleting the corresponding genomic region of one allele of each gene (*Figure 5*, *Figure 5—figure supplement 1*). The resulting cell lines (Casp9ΔE3-6(-/+), BimΔE2b-2c(-/+), BaxΔE3(-/+)) allow us to alter the isoform abundance at basal expression levels rather than through overexpression of cDNA. Moreover, the use of heterozygous deletions allow us to more closely mimic the partial isoform abundance changes that are induced upon T cell stimulation, rather than forcing complete ablation of a particular isoform. The deletion of the alternative exon in each cell clone was validated though genomic PCR and subjected to Sanger sequencing (*Figure 5—figure supplements 1 and 2*), and the impact on RNA and protein isoform expression was confirmed by RT-PCR and Western, respectively (*Figure 5*, *Figure 5—figure*

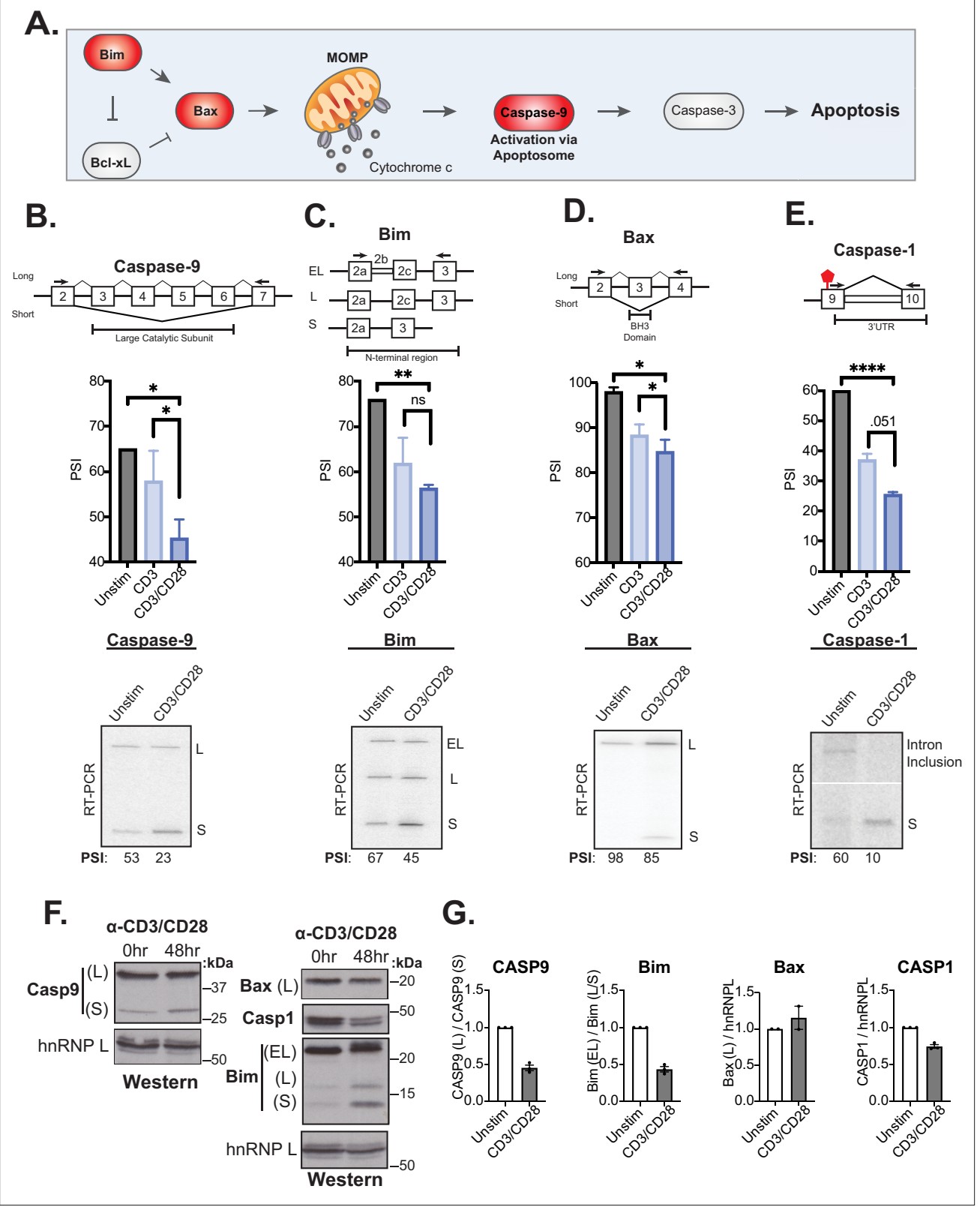

**Figure 4.** Splicing events enhanced by CD28 costimulation include those in genes involved in controlling apoptosis. (**A**) Schematic of the intrinsic apoptosis signaling pathway highlights in red the role of Bim, Bax, and Caspase-9 in regulating apoptosis. (**B–E**) Primary human naïve (CD4+ CD45RO-) T cells from three human donors were stimulated for 48 hr with either anti-CD3 or anti-CD3CD28 antibodies. After RNA was harvested, low-cycle radioactive RT-PCR analysis of splicing changes were averaged, as shown in bar graphs (n=3). Arrows on splicing diagrams represent the proximal

*Figure 4 continued on next page*

*Figure 4 continued*

location of primers on the transcript which flanks sequences regulated by alternative splicing for each gene. A representative low-cycle RT-PCR gel from one donor displays splicing isoforms for each gene and associated quantified percent spliced isoform (PSI) values under each gel. Significance was assessed by Student's t test and mean ± SEM values are plotted; p<0.05 (*), p<0.01(**), p<0.001(***). (**F**) Proteins were extracted from unstimulated and 48 hr-CD3CD28 stimulated primary human CD4+ T cells from one representative donor and blotted for caspase-9, Bax, caspase-1, Bim protein levels. hnRNPL is used as a loading control. (**G**) Bar graph quantifying average protein isoform levels with mean ± SEM values plotted. Each dot represents one human donor.

The online version of this article includes the following source data and figure supplement(s) for figure 4:

**Source data 1.** Full gels for *Figure 4B* RT-PCR analysis of CASP9 splicing.

**Source data 2.** Full gels for *Figure 4C* RT-PCR analysis of Bim splicing.

**Source data 3.** Full gels for *Figure 4D* RT-PCR analysis of Bax splicing.

**Source data 4.** Full gels for *Figure 4E* RT-PCR analysis of CASP1 splicing.

**Source data 5.** Full gels for *Figure 4F* Protein expression analysis.

**Figure supplement 1.** Further analysis of alternative splicing (AS) of apoptotic-related genes.

*supplement 1*). To control for off-target effects, we also analyzed the expression of other canonical apoptosis signaling proteins to confirm there were no confounding changes in other apoptotic proteins (*Figure 5—figure supplement 1*).

Using these isoform-modified cell lines, we then induced apoptosis signaling to ask if altering the isoform expression of caspase-9, Bim, or Bax conferred any changes in cell survival. Each cell line was incubated with titrating amounts etoposide or camptothecin for 6 hr, with an additional late timepoint of 24 hr. Camptothecin and etoposide are both DNA topoisomerase inhibitors that induce DNA damage and are well characterized to induce apoptosis signaling (*Ferraro et al., 2000*). Strikingly, the Casp9ΔE3-6(-/+) cell line presented robust resistance to apoptosis induction compared to wildtype cells in the presence of etoposide and camptothecin (*Figure 5B and C*). BimΔE2b-2c(-/+) and BaxΔE3(-/+) also exhibited resistance to apoptosis induction with camptothecin and etoposide (*Figure 5E, F, I, J*, *Figure 5—figure supplement 1*); however, the effects were not as large compared to Casp9ΔE3-6(-/+). As a complementary method to specifically confirm inhibition of apoptosis as the mechanism by which the Casp9ΔE3-6(-/+), BimΔE2b-2c(-/+), and BaxΔE3(-/+) cell lines exhibit reduced cell death, we quantified the levels of caspase-3/7 cleavage and MOMP activity in the presence of camptothecin and etoposide inhibitors. Similar to trends presented in overall cellular survival analysis, the levels of caspase-3/7 activity and MOMP activity were also decreased in all the modified cell line in the presence of camptothecin and etoposide compared to wildtype cells (*Figure 5—figure supplement 3*). In summary, increased abundance of the shorter isoform of caspase-9, Bim, and Bax increases cell survival upon apoptotic induction with camptothecin and etoposide.

The above experiments with the modified cell lines test the individual contribution of modulating isoforms of caspase-9, Bax, and Bim on apoptosis; however, upon activation of primary CD4+ T cells, the splicing of all three of these genes occurs in concert (*Figure 4*). We therefore wanted to ask if the increased abundance of the shorter isoform of caspase-9, Bax, and Bim produce additive effects to regulate cell survival. To this end, we transfected Jurkats with antisense morpholino oligos (AMOs) to block splice sites near caspase-9, Bim, and Bax alternative splicing events. Upon transfection in Jurkats cells, the longer isoform of caspase-9, Bim, Bax, decreased in abundance, as quantified by RT-PCR (*Figure 6A*). The decreased abundance of the longer isoform of caspase-9, Bim, and Bax is reflective of the trend seen upon primary CD4+ T cell activation and in the CRISPR-modified cell lines engineered to express less of the longer isoforms. As a negative control, Jurkats were also transfected with an AMO targeting an alternative exon within MKK7, which we have characterized previously (*Martinez et al., 2015*). We also controlled for specificity and off-target effects between AMOs by quantifying splicing of genes that the AMOs are not expected to regulate and observed no significant cross-regulation (*Figure 6—figure supplement 1A*).

Jurkat cells transfected with AMOs were treated with etoposide and camptothecin for 6 hr and apoptosis was again quantified by assessing the percentages of early dying cells (Annexin V+, PI- cells) by flow cytometry analysis (*Figure 6B–C*). Notably, when Jurkat T cells are transfected with all three AMOs (caspase-9, Bim, Bax) at once, there is a significant decreased fold change (FC) in early dying cells compared to cells individually transfected with caspase-9, Bax, and Bim AMOs. By contrast,

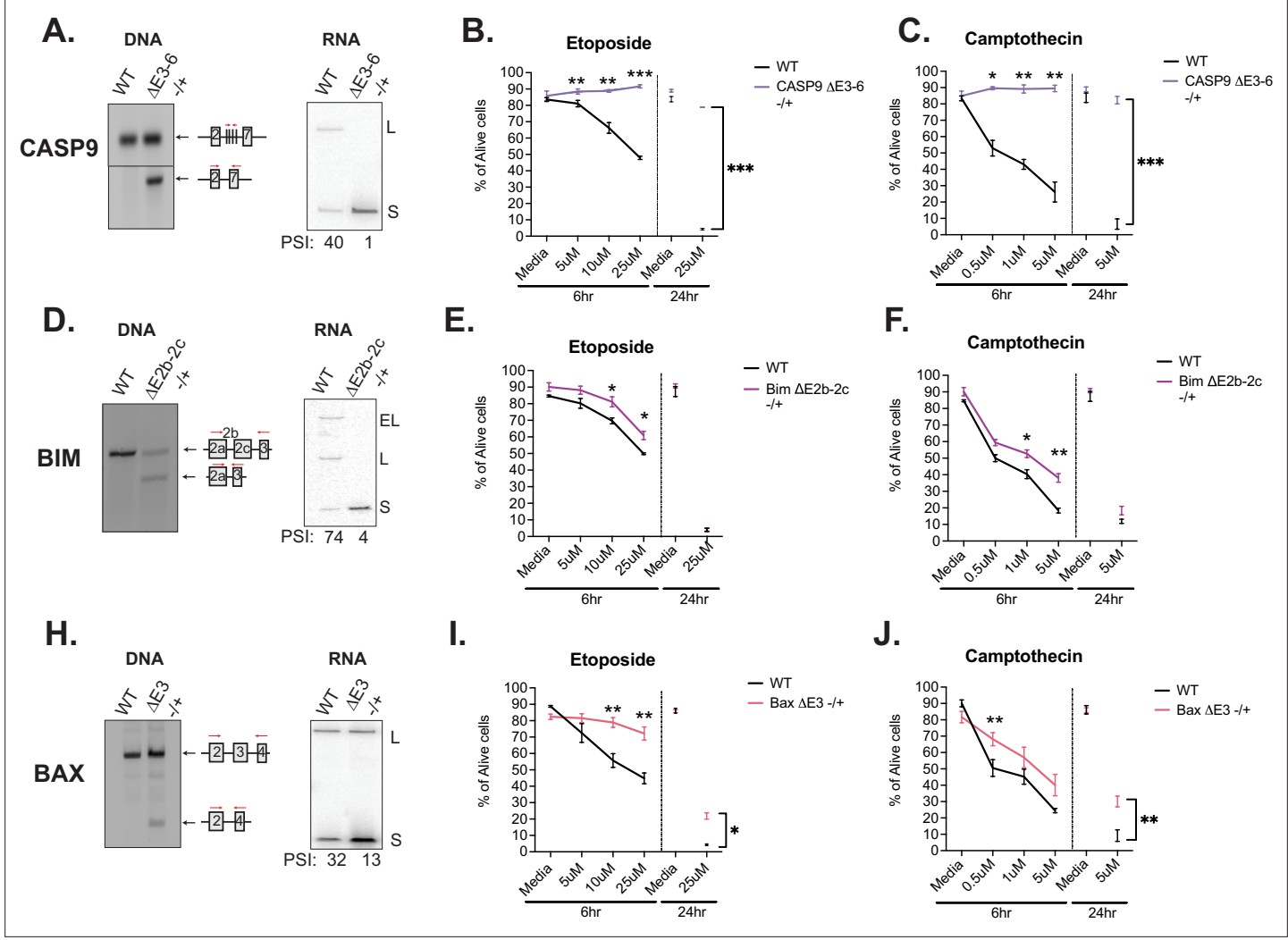

**Figure 5.** Activation-induced alternative splicing of apoptotic genes promotes T cell survival. (**A**) Casp9ΔE3-6-/+ is a Jurkat derived monoclonal cell line that expresses decreased levels of the longer isoform of caspase-9 by heterozygous CRISPR/Cas9 targeting and deletion of exons 3–6. Editing was confirmed at the DNA level by genomic PCR amplification. At the RNA level, low-cycle RT-PCR confirms caspase-9 isoform expression changes. Casp9ΔE3-6-/+ cells were treated with either etoposide (**B**) or camptothecin (**C**) to induce apoptosis at the specified concentrations and timepoints. The percentage of alive cells were quantified through flow cytometry analysis by Annexin V and PI staining. Values differing from wildtype control across three independent experiments are noted. Significance was measured by Student's t test with mean ± SEM values depicted; p<0.05 (*), p<0.01(**), p<0.001(***). (**D–J**) The validation and measurement of survival rates of BimΔE2b-2c-/+ and BaxΔE3-/+ cell lines are similar as described in **A-C**. Data from independent heterozygous clones is shown in *Figure 5—figure supplement 1*.

The online version of this article includes the following source data and figure supplement(s) for figure 5:

**Source data 1.** Full gels for *Figure 5A* PCR and RT-PCR analysis of CASP9 CRISPR cell line.

**Source data 2.** Full gels for *Figure 5D* PCR and RT-PCR analysis of Bim CRISPR cell line.

**Source data 3.** Full gels for *Figure 5H* PCR and RT-PCR analysis of Bax CRISPR cell line.

**Figure supplement 1.** Further data validating CRISPR clones.

**Figure supplement 1—source data 1.** Full gels for *Figure 5—figure supplement 1* protein expression analysis of CRISPR cell lines.

**Figure supplement 2.** Sequence of wildtype allele in CRISPR heterozygous clones.

**Figure supplement 3.** Assessment of additional apoptotic markers in CRISPR clones.

cells transfected with 10 nmol of MKK7 AMOs, to reflect the concentrations of AMOs individually transfected into cells, or 30 nmol, which reflects the final concentration of three AMOs transfected together, did not have a significant impact on cell death (*Figure 6B and C*). To further test the model that the enhancement of splicing induced by CD28 is important to reach a threshold for resistance

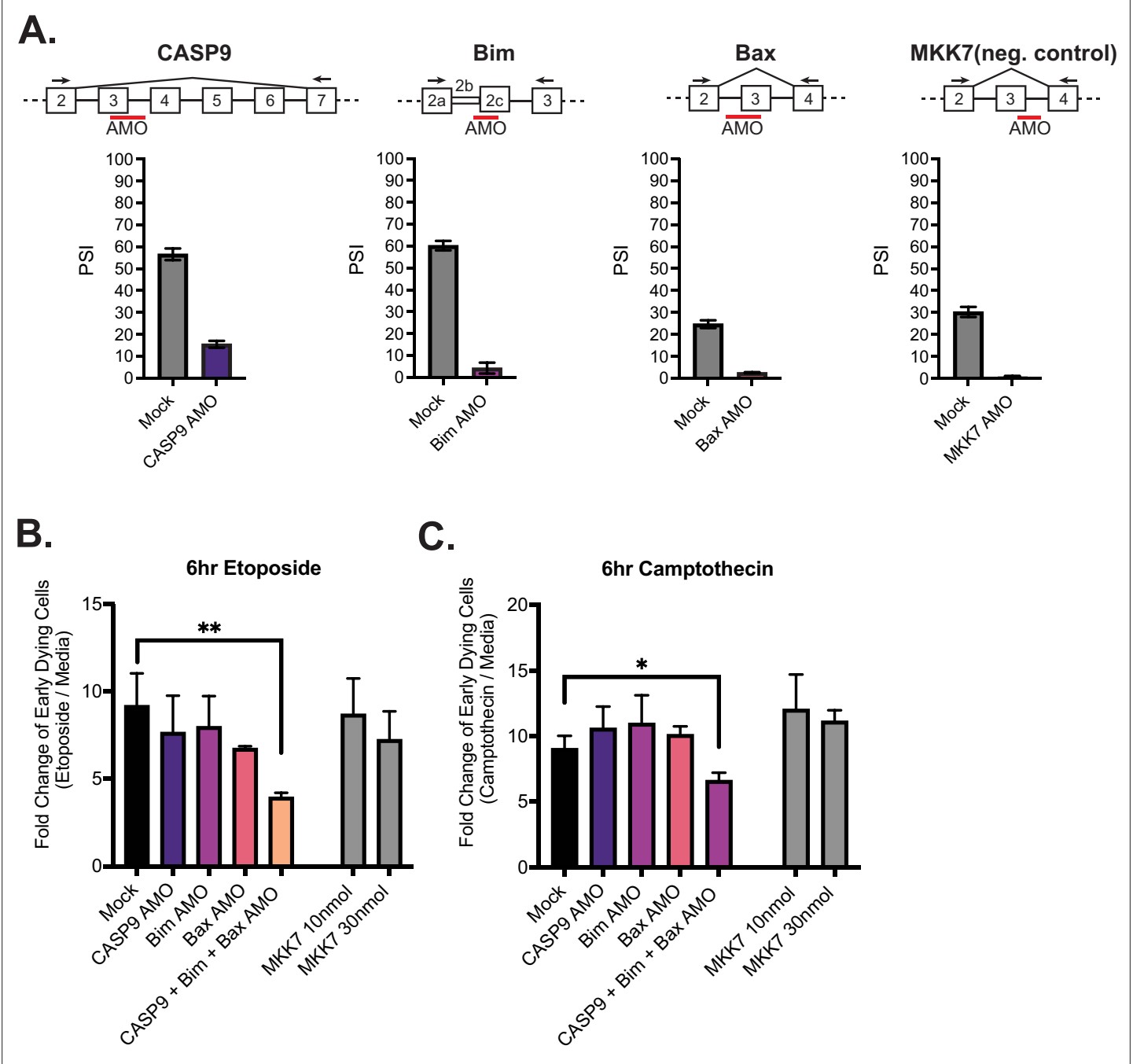

**Figure 6.** Activation-induced alternative splicing of apoptotic genes promotes T cell survival. (**A**) Jurkats were individually transfected with antisense morpholino oligos (AMOs) that targeted splicing events within caspase-9, Bim, Bax, and MKK7 (negative control) genes, and rested for 48 hr, then subjected to low-cycle RT-PCR analysis to quantify splicing changes within each gene. Splicing diagrams represent relative placement of AMO along transcripts of each gene. n=3 independent experiments. As shown previously (*Goehe et al., 2010*), blocking of caspase-9 exon 3 is sufficient to cause skipping of the whole exon 3–6 cassette. (**B**) Forty-eight hr after AMO transfection, cells were incubated with either etoposide or camptothecin for 6 hr and cell survival was quantified through staining of Annexin V and PI by flow cytometry. Early dying cells are characterized as Annexin V+ and PI-cells. Significance was measured by Student's t test across n=3 independent experiments with mean ± SEM values reported; p<0.05 (*), p<0.01(**). No differences between mock and single AMO treatments are statistically significant.

The online version of this article includes the following figure supplement(s) for figure 6:

**Figure supplement 1.** Additional analysis of antisense morpholino oligo (AMO)-treated Jurkat cells.

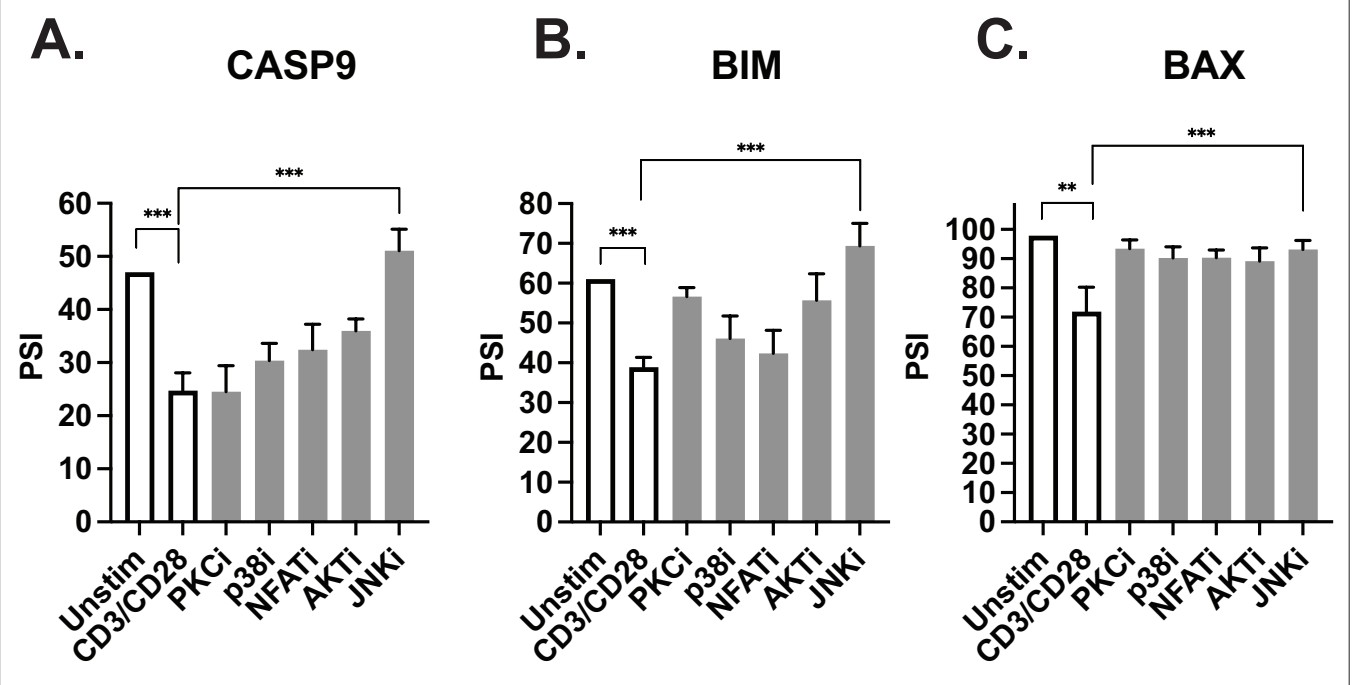

**Figure 7.** JNK signaling coordinates regulation of caspase-9, Bim, and Bax. (**A–C**) Primary CD4+ CD45RO- T cells isolated from three human donors were individually incubated with inhibitors for 1 hr and then stimulated with anti-CD3CD28 antibodies for 48 hr. The effect of each signal transduction target was averaged between at least two different inhibitors (see Materials and methods). Bar graphs represent low-cycle radioactive RT-PCR analysis of splicing changes within casapse-9, Bim, and Bax genes, as described in **Figure 4**. Significance was measured by Student's t test with mean ± SEM values reported; p<0.05 (*), p<0.01(**), p<0.001(***). n=3, independent experiments.

to apoptosis, we repeated the AMO experiment with a lower dose of oligomer to achieve a smaller change (two- threefold smaller dPSI) in splicing (**Figure 6—figure supplement 1B**). Strikingly, this reduced change in splicing was insufficient to yield any individual or additive effect on resistance to apoptosis (**Figure 6—figure supplement 1B**).

While it is difficult to draw a direct one-to-one comparison between the absolute change in splicing and resistance to apoptosis in primary T cells versus Jurkat cells due to general differences in cell physiology, these data provide strong support for a threshold effect between alternative splicing and apoptosis. Specifically, the CRISPR mutants demonstrate that robust changes in splicing of caspase-9, Bim, and Bax lead to resistance to apoptosis; while the low-dose AMO data demonstrate that individual or small changes in the splicing of caspase-9, Bim, and Bax have little impact on apoptosis. Only once a threshold of splicing change is reached in multiple genes is resistance to apoptosis observed. In sum, we conclude that coordinated and robust removal of critical domains in caspase-9, Bim, and Bax by alternative splicing, as observed upon activation of T cells by CD3 and CD28, is required to achieve a combinatorial enhancement of cell survival.

## JNK signaling coordinates regulation of caspase-9, Bim, and Bax

Previous studies in our lab have focused on identifying specific signaling pathways downstream of TCR stimulation that regulate splicing changes within T cells (**Heyd and Lynch, 2010**; **Martinez et al., 2015**). Interestingly, one of these studies demonstrated that Bax alternative splicing in primary CD4+ T cells is dependent on the JNK signaling pathway (**Martinez et al., 2015**). Moreover, JNK signaling in T cells has been shown to be enhanced by costimulation of CD28, compared to CD3 alone (**Su et al., 1994**). Thus, the requirement for JNK signaling is consistent with the enhancement of alternative splicing of Bax we observe upon CD28 costimulation. Consistent with the coordinated regulation of alternative splicing of apoptotic genes, we find that JNK activity is also required for the alternative splicing of caspase-9 and Bim upon T cell activation, while inhibition of other kinases have a weaker or more inconsistent effect on splicing of these genes (**Figure 7A–C**).

In our prior analysis of JNK-responsive splicing events, we found the region around JNK-sensitive exons are enriched for motifs corresponding to the binding sites for CELF2, PTBP1, and SRSF5 (*Martinez et al., 2015*). The abundance of CELF2 does not change appreciably upon stimulation of CD4+ T cells, however PTBP1 mRNA increases threefold upon CD3 or CD3/CD28 stimulation, while the mRNA encoding SRSF5 decreases twofold under these conditions (*Supplementary file 1*). More-over, motifs consistent with the binding sites for CELF2 (CCTGCC/GCCTGG), PTBP1 (CTTCTCC), and SRSF5 (AGGCAGAA) are enriched in or around exons that are enhanced by CD28 costimulation at 48 hr (*Supplementary file 4*), which include caspase-9, Bim, and Bax. Therefore, we predict that at least a subset of the splicing regulatory proteins CELF2, PTBP1, and SRSF5 coordinately control apoptotic-related splicing events in T cells. However, as we are not able to deplete these proteins in primary CD4+ T cells, and the Jurkat cell line does not recapitulate activation-induced splicing of the apoptotic genes, additional model systems will need to be developed to fully confirm this prediction.

## Discussion

Changes in alternative splicing induced upon T cell stimulation represent a dynamic genetic regulatory process that influences T cell biology and effector responses. Notably, our lab and others have found that approximately 10–15% of alternatively spliced genes undergo isoform abundance changes upon T cell activation. Here, we demonstrate that 10–20% of these changes are further enhanced by CD28 costimulation, with CD28 exhibiting a greater impact on splicing changes induced rapidly after stimulation (8 hr) as compared to later (48 hr). Critically, we show that a subset of splicing events influenced by the presence of CD28 costimulation occur in genes encoding the apoptosis signaling mediators, caspase-9, Bim, and Bax, and the isoforms upregulated by CD28 costimulation within these genes promote T cell survival upon stimulation. Together, these data indicate that CD28 costimulation regulates a network of alternative splicing events within the apoptosis signaling pathway to increase cell survival upon T cell stimulation.

Our analysis of the temporal and CD28 dependence of alternative splicing regulation downstream of T cell activation mirrors previous findings regarding transcriptional control. Specifically, for both splicing and gene expression, we find a greater extent of changes at later timepoints following activation as opposed to early. At least for splicing, genes which display different patterns of temporal regulation also exhibit differential GO enrichment analysis, suggesting that splicing events are regulated temporally so as to regulate different aspects of T cell biology and effector functions. Moreover, CD28 costimulation largely influences quantitative rather than qualitative changes in transcription and splicing; with the greatest impact in both cases observed at early timepoints following activation (*Figure 2*; *Diehn et al., 2002*; *Riley et al., 2002*). In other words, CD28 costimulation does not broadly lead to changes in the identity of genes undergoing alternative splicing or transcriptional activation, rather enhances the changes induced by signaling through the TCR, especially early in the time course. This impact of CD28 on gene expression is consistent with the fact that CD28 engagement isn't absolutely required for most signaling pathways downstream of T cell activation, rather enhances the degree to which several signaling pathways are induced. Importantly, however, despite similar patterns of regulation, the overlap of genes significantly regulated by differential expression and genes significantly regulated by splicing display a very minimal overlap. Therefore, splicing and transcription regulate distinct gene populations, a finding which further illustrates the need to specifically study splicing changes in T cells in order to understand the full complement of gene and protein expression changes that accompany and drive immune responses.

Given that CD28 costimulation regulates T cell survival, cytokine production, and proliferation, we were particularly excited to find that one of the most strongly CD28-induced splicing changes occurs in the apoptotic regulator, caspase-9. In addition, genes encoding several other apoptosis-related proteins, namely Bim, Bax, and Casp1, also exhibited CD28-enhanced splicing. Several studies have demonstrated that death signaling is regulated by changes in isoform expression due to alternative splicing, particularly in Fas/CD95 and Bcl-x in T cells and caspase-9 in cancer (*Cheng et al., 1994*; *Goehe et al., 2010*; *Izquierdo et al., 2005*; *Shultz et al., 2010*). However, the role of alternative splicing of Bim and Bax has not been confirmed, nor has caspase-9 alternative splicing been shown

to influence apoptosis in T cells. Moreover, a large caveat of previous studies is that the phenotypic roles of the differential isoforms were experimentally validated through artificial expression systems, despite ample evidence that overexpression of isoforms within apoptosis signaling genes can lead to confounding results. For example, basal versus overexpression of the long isoform of cFLIP, cFLIP(L), correlates with a differential apoptotic phenotype within cells (*Tsuchiya et al., 2015*). Therefore, in this study we sought to test the apoptotic function of the spliced isoforms at basal expression levels and ratios, by using heterozygous CRISPR editing and splice-blocking oligonucleotides.

Strikingly, these studies reveal that the smaller isoforms of caspase-9, Bim, and Bax, which are upregulated upon T cell stimulation, display an anti-apoptotic phenotype in Jurkat T cells (*Figure 5*). From a mechanistic point of view, the impact of splicing on caspase-9 function we observe is consistent with previous studies demonstrating that removal of the catalytic domain with skipping of exons 3–6 results in a dominant negative form of the protein (*Goehe et al., 2010*; *Seol and Billiar, 1999*; *Shultz et al., 2010*). Similarly, skipping of exon 3 of Bax removes the BH3 domain, which is essential for the homo- and hetero-dimerization of Bax to induce permeabilization of mitochondrial membranes (*Billen et al., 2008*; *Diaz et al., 1997*). Therefore, the short isoform of Bax cannot promote apoptosis. By contrast, how splicing of Bim leads to protection from apoptosis in T cells is less clear. Indeed, overexpression studies in non-hematopoietic cells has suggested that the smaller isoforms of Bim are more potent inducers of apoptosis (*O'Connor et al., 1998*). However, given that regulators of apoptosis have been shown to bind to the N-terminus of Bim (*Luciano et al., 2005*), which is the part of the protein regulated by splicing, we consider it likely that the smaller isoforms of Bim expressed at their endogenous level alter the stoichiometry of interaction between BCL-family members in T cells in a manner that blocks or fails to promote apoptosis.

Importantly, in addition to the individual contributions of alternative splicing of caspase-9, Bim, and Bax to controlling cell survival, we also find these splicing events work in combination to further enhance the anti-apoptotic phenotype when modulated simultaneously (*Figure 6*). We find that a commonality between these splicing events is the dependency on JNK signaling following costimulation. However, we note that each splicing event is also regulated by other signaling pathways downstream of TCR signaling, and that there is extensive cross-talk between signaling pathways and RBPs in T cells (*Martinez and Lynch, 2013*). Therefore, the full spectrum of events that may play a role in regulating caspase-9, Bim, and Bax AS events needs further elucidation. Since the alternative splicing of caspase-9, Bax, and Bim occur in the same temporal and JNK-dependent manner following costimulation of primary T cells stimulation at 48 hr, we conclude that these splicing events form a network to enhance T cell survival in primary human CD4+ T cells. Moreover, the fact that these splicing events are also all enhanced by the presence of CD28 costimulation demonstrates that CD28 costimulation increases cell survival and prevents anergy, not only through enhanced expression of IL2 and Bcl-xL as previously described (*Esensten et al., 2016*), but also through synergistic regulation of alternative splicing.

Given the typical combinatorial nature of splicing regulation, we predict that multiple RBPs and mechanisms likely contribute to the temporal and CD28-enhanced regulation of splicing during the activation of primary CD4+ T cells. Consistent with this prediction, we identified numerous RBPs that exhibit differential expression in a temporal and/or CD28-dependent manner (*Supplementary file 1*). Moreover, for a subset of these RBPs we find enriched binding motifs around regulated splicing events (*Supplementary file 4*; *Figure 2*). Future studies will be needed to disentangle the complex network of signaling events and RBPs that regulate specific subsets of genes during T cell activation.

In summary, we identify here differential temporal trends in alternative splicing between early and late T cell activation. In addition, we define a subset of splicing events that are enhanced by CD28 costimulation, including in the genes encoding caspase-9, Bax, Bim, and Casp1, and identify the signaling pathways and RBPs that may coordinate these CD28-enhanced splicing events. Most importantly, our data demonstrate the functional significance of CD28-enhanced splicing changes, as the resulting changes in expression of the apoptosis signaling proteins caspase-9, Bax, and Bim induced by splicing promote T cell survival upon induction of apoptosis signaling, and likely contribute to the CD28-induced survival of primary CD4+ T cells upon T cell activation.

# Materials and methods

**Key resources table**

| Reagent type (species) or resource | Designation | Source or reference | Identifiers | Additional information |
|---|---|---|---|---|
| Gene (*Homo sapiens*) | CASP9 | GenBank | #842 | |
| Gene (*Homo sapiens*) | BCL2L11 | GenBank | #10018 | |
| Gene (*Homo sapiens*) | BAX | GenBank | #581 | |
| Gene (*Homo sapiens*) | CASP1 | GenBank | #834 | |
| Cell line (*Homo sapiens*) | Wildtype Jurkat T cells | *Lynch and Weiss, 2000* | JSL1 | Clonal population of Jurkat T cells |
| Cell line (*Homo sapiens*) | Casp9ΔE3-6 (-/+) | This paper | | Cell line generated by CRISPR/CAS9 system |
| Cell line (*Homo sapiens*) | BimΔE2b-2c (-/+) | This paper | | Cell line generated by CRISPR/CAS9 system |
| Cell line (*Homo sapiens*) | and BaxΔE3 (-/+) | This paper | | Cell line generated by CRISPR/CAS9 system |
| Transfected construct (*Homo sapiens*) | pSpCas9(BB)–2A-GFP (PX458) | Addgene | Plasmid #48138 | Construct to transfect cells with Crispr guide RNAs |
| Biological sample (*Homo sapiens*) | Primary Human CD4+ T cells | Human Immunology core at the University of Pennsylvania | | CD4+ T cells enriched from apheresis product |
| Antibody | Anti-human CD69-PE (Mouse monoclonal) | Biolegend | Cat # 310905 | 1:100 |
| Antibody | Anti-human CD28 L293 (Mouse monoclonal) | BD | Cat # 348040 | 2.5 µg/mL of soluble antibody |
| Antibody | Anti-human CD3 NA/LE (Mouse monoclonal) | BD | Cat # 555336 | 2.5 µg/mL of plate bound antibody |
| Antibody | Anti-human Caspase-9 (Rabbit polyclonal) | Cell Signaling | Cat # 9502 | 1:1000 |
| Antibody | Anti-human Bim (Rabbit monoclonal) | Abcam | Cat # ab32158 | 1:1000 |
| Antibody | Anti-human Bax (Rabbit monoclonal) | Abcam | Cat # ab325034 | 1:1000 |
| Antibody | Anti-human Caspase-1 (Rabbit monoclonal) | Abcam | Cat # ab179515 | 1:1000 |
| Peptide, recombinant protein | Annexin V- Pacific Blue | Biolegend | Cat # 640917 | |
| Peptide, recombinant protein | IL-2 | Miltenyi | Cat # 130-097-742 | |
| Commercial assay or kit | RNA STAT-60 | Tel-Test Inc | | |
| Commercial assay or kit | Anti-human CD45RO human microbeads | Biolegend | Cat # 130-046-001 | |
| Commercial assay or kit | Caspase-3/7 FAM-FLICA assay | Immunochemistry Technologies | Cat # 93 | |
| Chemical compound, drug | Camptothecin | Millipore Sigma | Cat # 208925 | |
| Chemical compound, drug | Etoposide | Sigma-Aldrich | Cat # E1383 | |
| Chemical compound, drug | Propidium Iodide | Thermo Fisher | Cat # c34557 | |
| Chemical compound, drug | MitoTracker Red CMXRos | Thermo Fisher | Cat # M7512 | |
| Chemical compound, drug | R0-31-8220 | Selleckchem | S7207 | |
| Chemical compound, drug | Go6983 | Selleckchem | S2911 | |

*Continued on next page*

*Continued*

| Reagent type (species) or resource | Designation | Source or reference | Identifiers | Additional information |
|---|---|---|---|---|
| Chemical compound, drug | SB20350 | Selleckchem | S1076 | |
| Chemical compound, drug | Skepinone-L | Selleckchem | S7214 | |
| Chemical compound, drug | Cyclosporin A | Selleckchem | S2286 | |
| Chemical compound, drug | FK506 | Selleckchem | S5003 | |
| Chemical compound, drug | MK-2206 | Selleckchem | S1078 | |
| Chemical compound, drug | SP600215 | Selleckchem | S1460 | |
| Chemical compound, drug | JNK-IN-8 | Selleckchem | S4901 | |
| Chemical compound, drug | Tanzisertib | Selleckchem | S8490 | |
| Software, algorithm | PANTHER Classification System | PANTHER Classification System | Version 17.0 | |
| Software, algorithm | STREME algorithm | MEME suite software | Version 5.4.1 | |
| Software, algorithm | Flowjo | Flowjo | Version 10.7.1 | |
| Software, algorithm | GraphPad Prism | GraphPad Software | Version 9.4.0 | |
| Software, algorithm | RStudio | The R foundation | Version 1.1.463 | |

## Isolation and stimulation of primary human CD4+ T cells and Jurkat cells

CD4+ T cells were obtained by apheresis from de-identified healthy blood donors after informed consent by the University of Pennsylvania Human Immunology Core. Samples were collected from three donors, Donor 1 (age: 46, sex: male), Donor 2 (age: 26, sex: female), and Donor 3 (age: 32, sex: male). Naïve CD4+ T cells were negatively enriched for by utilizing CD45RO microbeads (Miltenyi: 130-046-001). $5 \times 10^6$ naïve CD4+ T cells were stimulated in complete RPMI supplemented with 10 IU of IL-2 (Miltenyi: 130-097-742), with either soluble 2.5 µg/mL anti-CD28 (BD: 348040) or 2.5 µg/mL bound anti-CD3 (BD: 555336), or with both soluble anti-CD28 and bound anti-CD3. Cells were harvested after 8 and 48 hr of culture. Stimulation efficiency and consistency between human donors was measured by CD69 expression (Biolegend: 310905), analyzed by flow cytometry on the BD LSRII equipment maintained by the Flow Cytometry core at the University of Pennsylvania. Growth and transfection of Jurkat cells was as described previously (*Lynch and Weiss, 2000*). Jurkat cells were validated by RNA-sequencing and negative for mycoplasm.

## RNA-sequencing of cultured primary CD4+ T cells with alternative splicing and expression quantifications

RNA was isolated with RNA Bee (Tel-Test Inc), according to the manufacturer's protocol, from cultured primary human CD4+ T cells described above. The RNA integrity number (RIN) was measured with the Agilent bioanalyzer, and all samples had a RIN >8.0. RNA-sequencing libraries were generated by and sequenced by GeneWiz. The libraries were poly(A) selected (nonstranded) and paired-end sequenced at a 150 bp read length. Alternative splicing was quantified by MAJIQ and differential expression analysis was quantified by DESeq2. Significant alternative splicing events analyzed by MAJIQ have >10% dPSI and p<0.05 and significant differentially expressed genes are >1.5 log2FC (stimulated/unstimulated). For more, detailed information about the experimental design was published previously (*Radens et al., 2020*). The RNA-sequencing data generated for this study is available in GEO (GSE135118). For RBP expression analysis, list of RBPs originated from *Gerstberger et al., 2014*.

## GO enrichment analysis

GO enrichment analysis was performed using the GO biological process compete tool provided by the PANTHER Classification System (http://pantherdb.org). The splicing events categorized as 'Late/

Sustained Events' and 'Early/Sustained Events' (*Supplementary file 3*) were used over whole genome background control. The significant GO enrichment analysis met the threshold of p<0.05 and an enrichment score >2.

## Motif enrichment analysis

Motif enrichment analysis was provided by STREME algorithm in the MEME suite software v5.4.1 (https://meme-suite.org). For simplicity, we quantified sequences that are enriched from splicing events that are cassette exons. Sequences, in hg37 coordinates, selected from each alternative splicing event consisted of 250 bp downstream of the alternative cassette exon, the sequences encoding the cassette exon, and 250 bp upstream of the alternative cassette exon. Upstream sequences exclude 7 bp that encodes for the conserved 5' splice site and downstream sequences exclude 30 bp that involves the 3' splice site to avoid sequence enrichment. Significant enriched motifs listed met the threshold of E-value <0.05 and enrichment ratio >2 (*Supplementary file 3*). Candidate RBPs predicted to bind to the enriched motifs were determined through annotated binding specificities provided by two databases (http://cisbp-rna.ccbr.utoronto.ca, https://attract.cnic.es/index).

## Validation of alternative splicing changes by RT-PCR

RNA was isolated from primary CD4+ T cells or Jurkats using RNABee or RNAStat (Tel-Test, Inc) according to the manufacturer's protocol. Low-cycle radioactive RT-PCR was performed and analyzed as previously described in detail (*Ip et al., 2007*; *Lynch and Weiss, 2000*; *Rothrock et al., 2003*). Primer sequences used for splicing analysis include: caspase-9 (F: GGCCAGGCAGCTGATCATAG ATCTG R: GGAGGCCACCTCAAACCCATGGTC), Bim (F: CCTTCTGATGTAAGTTCTGAGTGTGAC R: CCATATCTCTGGGCGCATATCTGC), Bax (F: CTCTGAGCAGATCATGAAGACAGG R: GAAAACAT GTCAGCTGCCACTCG), and Caspase-1 (F: GAGCAGCCAGATGGTAGAGCGCAG R: CCTTTACA GAAGGATCTCTTCACTTC).

## Western blotting

Ten µg of total protein lysates were loaded into 10% 37.5:1 bis-acrylamide SDS-PAGE gels. SDS-PAGE gels were transferred onto PDVF membranes and targeted proteins were visualized with a chemiluminescence system and subsequent imaging with an x-ray developer. Antibodies used to detect protein expression levels are as follows: caspase-9 (Cell Signaling: 9502), Bim (Abcam:32158), Bax (Abcam: ab325034), and Caspase-1 (Abcam: ab179515).

## Identifying cell survival upon apoptosis induction monoclonal cell lines via Crispr/Cas9 system

To generate individual cell clones that express decreased levels of the longer isoform of caspase-9, Bim, and Bax, we used CRISPR/Cas9 targeted editing in Jurkat cells to delete the genomic elements that encode for the regulated alternative splicing events. Two custom gRNAs were created to target intronic regions that surround genomic elements that encode for the alternative splicing event of interest. The gRNAs designed sequences are as follows: caspase-9 (gRNA-1: CTTTGATATATACCTAAGGG, gRNA-2: GTGGCCACAGCTAAACTGCA), Bim (gRNA-1: CACTGGAGGATCGAGACAGC, gRNA-2: GGTAAGAG GCAGTTGACGTG), and Bax (gRNA-1: GAGGGTGCAGAATCAGAACG, gRNA-2: GAAAAGCAACAG GCCAACGG). gRNAs were individually cloned into a BBSP1 digested PX458-462 plasmid (https://www.addgene.org/48138) which also encodes for both the Cas9 enzyme and GFP.

Plasmids with cloned-in gRNA sequences that target upstream and downstream intronic regions flanking the splicing event of interest were combined and transfected (5 µg each) into 10 million Jurkat cells through electroporation using the Bio-Rad Gene Pulser Xcell. After electroporation, edited Jurkat cells were rested for 48 hr and sorted into 96-well plates (FACS Jazz) by gating on doublet exclusion and GFP+ expression. One or two to three cells were added per well for the growth of single cell colonies. Individual cell clones were screened by PCR amplification of targeted genomic regions for band size exclusion on an 1.5% EtBr agarose gel, sequencing of genomic DNA, and by Western blot. All Jurkat cells were cultured in RPMI (Corning) supplemented with 5% heat-inactivated fetal bovine serum (FBS) (GIBCO) as described previously (*Lynch and Weiss, 2000*). Cells were incubated with apoptotic inducers camptothecin (MilliporeSigma: 208925) and etoposide (Sigma-Aldrich: E1383) for the specified concentrations and timepoints for downstream flow cytometry analysis.

Cell survival analysis was measured by flow cytometry through Annexin V-Pacific Blue (Biolegend: 640917) and Propidium Iodide staining (Thermo Fisher: c34557). Caspase-3/7 activity was measured with FAM-FLICA probes (Immunochemistry Technologies: 93) and MOMP events were measured with MitoTracker Red CMXRos staining (Thermo Fisher: M7512).

## Determining the synergistic effects of splicing events to promote cell survival in Jurkat T cells

For each AMO treatment, 10 million Jurkat cells cultured in RPMI media without FBS or antibiotics were electroporated using a Bio-Rad Gene Pulser Xcell. Ten nmol of each individual AMO were added in the transfection whether the cells were transfected with individual AMOs or in conditions where the AMOs are pooled together. Transfected cells were seeded into six-well plates with pre-warmed RPMI supplemented with 5% of FBS and rested for 48 hr. The splice-blocking activity of each AMO at 48 hr was validated by radio-labeled low-cycle RT-PCR using primer sequences for caspase-9, Bim, and Bax described above. Primer sequences used to validate the MKK7 splicing event include: F: CGACCTCA ACCTGGATATCAGCC, R: GGAGCTCTCTGAGGATGGCGAGC. In addition, 48 hr after transfection, cells were incubated with apoptotic inducers, 5 µM camptothecin and 25 µM etoposide, for 6 hr, and cell survival analysis was measured via flow cytometry with Annexin V-Pacific Blue and Propidium Iodide staining.

## Elucidation of signal transduction pathways downstream of TCR stimulation to regulate splicing changes in primary CD4+ T cells

To elucidate the roles of signal transduction proteins to regulate splicing changes, each protein known to be regulated downstream of TCR signaling was targeted by at least two different inhibitors. Human primary CD4+ CD45RO- T cells were incubated with individual inhibitors for 1 hr and stimulated with bound anti-CD3 and soluble anti-CD28 antibodies for 48 hr. Cells were then harvested for RNA purification and low-cycle RT-PCR analysis for splicing quantifications. The inhibitors utilized are as follows: PCKi: R0-31-8220 (Selleckchem: S7207) and Go6983 (Selleckchem: S2911), p38i: SB20350 (Selleckchem: S1076) and Skepinone-L (Selleckchem: S7214), NFATi: Cyclosporin A (Selleckchem: S2286) and FK506 (Selleckchem: S5003), AKTi: MK-2206 (Selleckchem:S1078) and Ipatasertib (Selleckchem:S2808), JNKi: SP600215 (Selleckchem: S1460) and JNK-IN-8 (Selleckchem: S4901) and Tanzisertib (Selleckchem: S8490).

## Acknowledgements

The authors acknowledge the support of NIH grants R35 GM118048 and GM118048-S1 (KWL) and F31 GM140978 (DB). We are grateful for the assistance of the Human Immunology Core and the Flow Cytometry Core at the University of Pennsylvania for isolation of the human CD4+ T cells (IRB # 811028) and flow cytometry analysis and sorting of Cas9-transfected cells, respectively.

## Additional information

### Competing interests

Kristen W Lynch: Reviewing editor, *eLife*. The other authors declare that no competing interests exist.

### Funding

| Funder | Grant reference number | Author |
|---|---|---|
| National Institute of General Medical Sciences | R35 GM118048 | Kristen W Lynch |
| National Institute of General Medical Sciences | F31 GM140978 | Davia Blake |

The funders had no role in study design, data collection and interpretation, or the decision to submit the work for publication.

## Author contributions
Davia Blake, Conceptualization, Data curation, Formal analysis, Funding acquisition, Validation, Investigation, Visualization, Methodology, Writing - original draft, Writing - review and editing; Caleb M Radens, Data curation, Software, Formal analysis; Max B Ferretti, Matthew R Gazzara, Software, Formal analysis, Writing - review and editing; Kristen W Lynch, Conceptualization, Supervision, Funding acquisition, Writing - original draft, Writing - review and editing

## Author ORCIDs
Kristen W Lynch ⓘD http://orcid.org/0000-0002-0120-8079

## Decision letter and Author response
Decision letter https://doi.org/10.7554/eLife.80953.sa1
Author response https://doi.org/10.7554/eLife.80953.sa2

# Additional files

## Supplementary files
• MDAR checklist

• Supplementary file 1. Differential gene expression changes upon CD3 and/or CD28 stimulation.

• Supplementary file 2. Alternative splicing events which are changed significantly upon CD3 and/or CD28 stimulation.

• Supplementary file 3. Changes in alternative splicing over time following CD3 and CD28 stimulation.

• Supplementary file 4. Motifs enriched around regulated alternative splicing events.

## Data availability
The RNA-seq data generated for this study is available in GEO under accession codes GSE135118.

The following previously published dataset was used:

| Author(s) | Year | Dataset title | Dataset URL | Database and Identifier |
|---|---|---|---|---|
| Davia B, Michael MJ, Caleb RM, Kristen LW | 2020 | CD28 regulation of global alternative splicing changes in activated human CD4+ T cells | https://www.ncbi.nlm.nih.gov/geo/query/acc.cgi?acc=GSE135118 | NCBI Gene Expression Omnibus, GSE135118 |

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
