## [Editor Report]

Blake and colleagues examine programs of alternative splicing in apoptotic proteins controlled during T-cell activation. Apoptotic regulators have long been known to often be expressed in pairs of pro- and anti-apoptotic isoforms. The demonstration of how a program of these splicing changes contributes to immune responses is significant to the understanding of both apoptosis and T-cell biology.

---

## [Decision Letter]

**Decision letter after peer review:**

Thank you for submitting your article "Alternative Splicing of Apoptosis Genes Promotes Human T Cell Survival" for consideration by *eLife*. Your article has been reviewed by 3 peer reviewers, one of whom is a member of our Board of Reviewing Ediotrs, and the evaluation has been overseen by Douglas Black as the Reviewing Editor and Tadatsugu Taniguchi as the Senior Editor. The reviewers have opted to remain anonymous.

Essential revisions:

Blake and colleagues examine programs of alternative splicing controlled during T cell activation. Using CD4^+^ T cells from human donors, cells were stimulated with anti-CD28, anti-CD3, and combined anti-CD3/28 antibodies. RNA was then isolated at 2 time points, sequenced, and analyzed for changes in spliced isoform ratios. T Cell Receptor stimulation alone via anti-CD3 is known to induce the anergic state resulting from suboptimal stimulation, while CD28 costimulation with CD3 induces many genes to a higher level of expression similar to stimulation by antigen-presenting cells. Analyzing the splicing responses to these stimuli, the authors find that CD28 costimulation also enhances the splicing changes that accompany T cell activation. A subset of these splicing targets encode apoptotic regulators including Caspase-9, Bax, and Bim. They show that forced expression of the isoforms that are increased by costimulation results in reduced apoptosis in Jurkat cells treated with apoptotic inducers. Using kinase inhibitor treatments they show that *Jnk* kinase activity is required for the splicing changes in the three apoptotic regulators. Apoptotic regulators have long been known to often be expressed in pairs of pro- and anti-apoptotic isoforms. The reviewers all agreed that demonstration of how a program of these splicing changes contributes to immune responses is significant to the understanding of both apoptosis and T cell biology. The results generally support the conclusions, although the extent to which the quantitative changes in splicing induced by CD28 co-stimulation contribute to anergy suppression is not fully clear.

Most important Issues.

1. In Figures 5A and D, the splicing changes for Casp9 and Bim appear to be substantially larger than expected for a heterozygous mutation. The longer isoforms coming from a wildtype allele are absent. The authors should assess whether the CRISPR protocol has not inactivated the second alleles in these cells. Ideally, they will be able to isolate additional clones that do express a 50:50 mix of isoforms. Alternatively, they will need to assess how the complete shift to one isoform affects their conclusions – see point 2 below.

2. Additional concerns come from the magnitude of the splicing changes induced by CD28 costimulation compared to the splicing changes that were induced in individual transcripts by Crispr and antisense oligos. The proposed model is that CD28 co-stimulation prevents anergy not only through enhanced expression of IL2 and BCl^-^xL, but also through synergistic regulation of alternative splicing. However, some alternative splicing events selected as candidate mediators of the biological effects also exhibit some change upon activation of CD3 alone. It is a question of whether the additional splicing changes induced by CD28 costimulation are sufficient to explain the CD28-mediated anergy suppression. For example, Figures 4B-E show examples in which co-stimulation by CD28 vs CD3 alone induces PSI changes of approx. 0.2 vs 0.07 (Caspase 9), 0.2 vs 0.15 (Bim, not statistically significant), 0.13 vs 0.10 (Bax) or 0.35 vs 0.22 (Caspase 1, close to statistical significance). When these events are modulated in Figures 5 and 6 to assess their biological effects, the PSI changes upon CRISPR gene editing or AMO treatment are typically much larger (e.g. 0.4 for Caspase 9, 0.7 for Bim). While it is clear that these alternative splicing events can influence T cell viability, it is less clear that the extra push induced on these splicing changes by CD28 co-stimulation makes a significant contribution to anergy suppression. Given the authors' model, they should consider titrating the AMOs to achieve changes in PSI similar to those observed upon CD28 co-stimulation and verify that they still collectively make a difference in cell viability.

Additional Points.

3. The initial analysis in Figure 1D would be strengthened by additional quantitative information about the distribution of alternative splicing changes. This might help explain why the overall numbers of altered splicing events do not change much for CD3/CD28 costimulation compared to CD3 alone. The authors set a threshold of >10% dPSI to be considered a significant event. It would be helpful to know how many of these start at 0-10 PSI prior to stimulation, how many start at 10-20 PSI, 20-30 PSI, etc., and how these binned distributions change between different stimuli. In Figure 1E the overlap between splicing changes across donor samples is rather limited; is the overlap higher if the threshold dPSI values are increased to consider larger numbers of splicing changes? The authors define the value PSI as "percent spliced isoform" but in the figures, it is not always clear which isoform is being measured. The term PSI was originally defined as "percent spliced in" and referred to one exon or segment of mRNA rather than an entire isoform, which might have several sites of alternative splicing. It may be that the analysis tools used by the authors define PSI by isoforms. There are some advantages to this but they need to then define the isoform being measured. It appears that it is usually an exon-included isoform, but is confusing because several of the RNAs being analyzed have multiple splicing events changing and are not just simple exon inclusion events.

4. Δ PSI values after CD3/CD28 costimulation are compared to those after CD3 stimulation alone by taking the ratio. In Figure 2EF, histograms are presented plotting numbers of genes against δ PSI ratios. This yields a peak at 1 indicating that most genes show no difference in their splicing change between the two stimuli. Of primary interest are genes showing greater splicing change after CD28 costimulation and these are seen trailing up from the peak to values ranging from 2 to 80 fold. Since the maximum δ PSI is 100%, the genes with high fold changes are presumably starting at very low PSI values to get an 80-fold change. It is not clear whether some of these large values are the result of taking the ratio of two very small numbers and are thus not indicative of a major change in PSI. From Figure 2AB, it appears that some are large changes in PSI but it should be made clearer how many changes are like this, perhaps by discussing how δ PSI cut-offs affect the observed changes. For example, they might examine the dPSI ratio distributions after separating the splicing events into bins based on the CD3 dPSI.

5. The authors should provide a list of the RNA binding proteins that are upregulated under various conditions. It is not in the referenced Supplemental Table 2. How were the enriched motifs connected to particular proteins? It appears they could be binding sites for several possible factors, besides the ones proposed. In Figure 2G, there appears to be a typo – do they mean hnRNP A1B2 or A2B1?

6. In Figure 3 it would be helpful to note the starting PSI value from unstimulated cells for each gene.

7. Figure 4F: A quantification of the changes in protein levels should be presented.

8. Figure 6: it is unclear why the AMO directed against the junction of exon 3 with intron 3 in CASP9 should cause skipping of all exons between 3 and 6 and thus lead to splicing between exons 2 and 7. It would be helpful to show an analysis of the patterns of splicing induced by this AMO in the region. More generally, the effects of combining AMOs in panels B and C are described as synergistic. It appears that the effects of combining AMOs with etoposide (panel B) are additive rather than synergistic. The combined AMOs also reduce camptothecin-mediated cell death, but puzzlingly the individual AMOs seem to stimulate the camptothecin effect (in contrast with the CRISPR results from Figure 5). These results need a better description.

9. In Figure 7, the exons all appear sensitive to *Jnk* inhibition, but their responses to other kinase inhibitors are quite distinct. The argument that they share a common regulatory pathway seems overstated. Given that the kinase inhibitor data are just an initial result and inconclusive, the authors should consider whether these data make a critical contribution to the manuscript or could be left out, pending a more thorough analysis.

---

## [Author Response]

Essential revisions:Blake and colleagues examine programs of alternative splicing controlled during T cell activation. Using CD4^+^ T cells from human donors, cells were stimulated with anti-CD28, anti-CD3, and combined anti-CD3/28 antibodies. RNA was then isolated at 2 time points, sequenced, and analyzed for changes in spliced isoform ratios. T Cell Receptor stimulation alone via anti-CD3 is known to induce the anergic state resulting from suboptimal stimulation, while CD28 costimulation with CD3 induces many genes to a higher level of expression similar to stimulation by antigen-presenting cells. Analyzing the splicing responses to these stimuli, the authors find that CD28 costimulation also enhances the splicing changes that accompany T cell activation. A subset of these splicing targets encode apoptotic regulators including Caspase-9, Bax, and Bim. They show that forced expression of the isoforms that are increased by costimulation results in reduced apoptosis in Jurkat cells treated with apoptotic inducers. Using kinase inhibitor treatments they show that Jnk kinase activity is required for the splicing changes in the three apoptotic regulators. Apoptotic regulators have long been known to often be expressed in pairs of pro- and anti-apoptotic isoforms. The reviewers all agreed that demonstration of how a program of these splicing changes contributes to immune responses is significant to the understanding of both apoptosis and T cell biology. The results generally support the conclusions, although the extent to which the quantitative changes in splicing induced by CD28 co-stimulation contribute to anergy suppression is not fully clear.

We are pleased the reviewers agree with the overall significance and importance of this study. As indicated in more depth below, we have provided more data and more clarification to support our model of a threshold effect in which robust changes in multiple splicing events, as occurs with CD28 costimulation, is required for robust resistance to apoptosis.

Most important Issues.1. In Figures 5A and D, the splicing changes for Casp9 and Bim appear to be substantially larger than expected for a heterozygous mutation. The longer isoforms coming from a wildtype allele are absent. The authors should assess whether the CRISPR protocol has not inactivated the second alleles in these cells.

To confirm that the CRISPR protocol has not inactivated the second allele we have sequences extensively around the splice sites of the WT allele. The results are provided in a new Supplemental Figure S7. The only clone in which we observe a small deletion upsteam of the exon 3 3’ss in the Bax heterogyzous, however, this is unlikely to impact splicing given the fact that we observe accurate exon 3 inclusion in this clone.

Ideally, they will be able to isolate additional clones that do express a 50:50 mix of isoforms. Alternatively, they will need to assess how the complete shift to one isoform affects their conclusions – see point 2 below.

We previously had isolated additional heterogyzous clones of Bim and Bax, which exhibit similar behavior to the clones shown in the main figure. Data from these second clones are shown in a revised Supplemental Figure 6. Although these still don’t have a 50:50 mix, the second Bim clone does express more of the long isoforms. Unfortunately, despite multiple attempts and the screening of a large number of clones we have not been able to isolate another CASP9 heterozygous clone, perhaps due to the larger deletion (~10 kb) and repair that is required for this gene editing. As detailed more below we have modified our description of the results and conclusions to clarify that our model is that splicing of these genes form a continuum of isoform expression that contributes to a continuum of survival. Consistent with this combinatorial model we also note that the resistance to survival of the CASP9 heterozygote is greater than that observed with the other clones; indeed almost 100%. Thus, if splicing was this extreme upon CD28 co-stimulation, no further additive effect of Bim and Bax AS would be needed. Finally, while we don’t have a clear explanation for why the CASP9 clone expresses so little of the full length isoform, the AMO experiment described below provides an important orthogonal assay to support our model.

2. Additional concerns come from the magnitude of the splicing changes induced by CD28 costimulation compared to the splicing changes that were induced in individual transcripts by Crispr and antisense oligos. The proposed model is that CD28 co-stimulation prevents anergy not only through enhanced expression of IL2 and BCl^-^xL, but also through synergistic regulation of alternative splicing. However, some alternative splicing events selected as candidate mediators of the biological effects also exhibit some change upon activation of CD3 alone. It is a question of whether the additional splicing changes induced by CD28 costimulation are sufficient to explain the CD28-mediated anergy suppression. For example, Figures 4B-E show examples in which co-stimulation by CD28 vs CD3 alone induces PSI changes of approx. 0.2 vs 0.07 (Caspase 9), 0.2 vs 0.15 (Bim, not statistically significant), 0.13 vs 0.10 (Bax) or 0.35 vs 0.22 (Caspase 1, close to statistical significance). When these events are modulated in Figures 5 and 6 to assess their biological effects, the PSI changes upon CRISPR gene editing or AMO treatment are typically much larger (e.g. 0.4 for Caspase 9, 0.7 for Bim). While it is clear that these alternative splicing events can influence T cell viability, it is less clear that the extra push induced on these splicing changes by CD28 co-stimulation makes a significant contribution to anergy suppression. Given the authors' model, they should consider titrating the AMOs to achieve changes in PSI similar to those observed upon CD28 co-stimulation and verify that they still collectively make a difference in cell viability.

We thank the authors for this excellent suggestion. As show in a new Supplemental Figure S9B, we have performed the suggested AMO titration experiment, and find that when the splicing change is reduced ~2-3 fold (similar to the difference in dPSI induced by CD28), we no longer observe any individual or additive impact on apoptosis. This threshold effect is consistent with the competitive nature of the pro- and anti- apoptotic forms of these genes induced by splicing, such that it is the balance of these two activities that can flip a switch to ultimately determine survival versus apoptosis.

We note that we would not expect a direct one-to-one relationship between the absolute dPSI or change in dPSI and resistance to apoptosis in the primary T cells versus Jurkat cells, due to general differences in cell physiology. Nevertheless, the data from the Jurkat cells provide strong support for (1) a correlation between the extent of alternative splicing and resistance to apoptosis as well as (2) a multiple-hit model in which altering several apoptotic pathway components promotes survival. These conclusions are both consistent with our proposed model that the increase in splicing induced by CD28 costimulation contributes to the overall resistance to apoptosis and anergy that is observed.

Additional Points.3. The initial analysis in Figure 1D would be strengthened by additional quantitative information about the distribution of alternative splicing changes. This might help explain why the overall numbers of altered splicing events do not change much for CD3/CD28 costimulation compared to CD3 alone. The authors set a threshold of >10% dPSI to be considered a significant event. It would be helpful to know how many of these start at 0-10 PSI prior to stimulation, how many start at 10-20 PSI, 20-30 PSI, etc., and how these binned distributions change between different stimuli.

We thank the reviewers for this useful suggestion to look deeper into our data. The results of the analysis suggested is now shown in a new Supplemental Figure S3. Importantly, while we see some bias in the starting PSI among changing events at 8 hours, there is no such bias at 48 hours. Therefore, we conclude that neither our analysis nor the biology of the splicing regulation can be explained by skewed PSI distributions that exist prior to stimulation.

In Figure 1E the overlap between splicing changes across donor samples is rather limited; is the overlap higher if the threshold dPSI values are increased to consider larger numbers of splicing changes?

While we understand the reviewer’s suggestion, we are hesitant to alter our dPSI threshold. Increasing the threshold would further limit the number of splicing changes, and likely reduce the overlap given the variability inherent to human subject. Decreasing the threshold (which we believe is what the reviewer meant) would certainly increase the number of changes and increase the overlap, but from previous studies we know this would also increase the percentage of false positives and decrease the validity of our results. We have now added clarification in the text for our choice of threshold. We also note that biological variability of splicing changes is expected between human donors, in accordance to published literature (e.g. Park et al., Am J Hum Genet 2018, Amoah et al., Genome Res 2021).

The authors define the value PSI as "percent spliced isoform" but in the figures, it is not always clear which isoform is being measured. The term PSI was originally defined as "percent spliced in" and referred to one exon or segment of mRNA rather than an entire isoform, which might have several sites of alternative splicing. It may be that the analysis tools used by the authors define PSI by isoforms. There are some advantages to this but they need to then define the isoform being measured. It appears that it is usually an exon-included isoform, but is confusing because several of the RNAs being analyzed have multiple splicing events changing and are not just simple exon inclusion events.

As the reviewers suspect, the MAJIQ algorithm defines PSI as Percent Spliced Isoform – which is typically the included form (to align with Percent Spliced In), but also accounts for complexities in splicing patterns. To improve clarity, we have revised Figure 3 to highlight the isoform that is being quantified.

4. Δ PSI values after CD3/CD28 costimulation are compared to those after CD3 stimulation alone by taking the ratio. In Figure 2EF, histograms are presented plotting numbers of genes against δ PSI ratios. This yields a peak at 1 indicating that most genes show no difference in their splicing change between the two stimuli. Of primary interest are genes showing greater splicing change after CD28 costimulation and these are seen trailing up from the peak to values ranging from 2 to 80 fold. Since the maximum δ PSI is 100%, the genes with high fold changes are presumably starting at very low PSI values to get an 80-fold change. It is not clear whether some of these large values are the result of taking the ratio of two very small numbers and are thus not indicative of a major change in PSI.

We appreciate this important concern from the reviewers. We, in fact, were also worried about the same issue when we did the initial analysis and thus used a pseudo-value in which all CD3 induced splicing events with a very low dPSI (< 1%) were normalized to a value of 1%. We realized that we forgot to make this important point clear in the initial version and have now added this to the text.

From Figure 2AB, it appears that some are large changes in PSI but it should be made clearer how many changes are like this, perhaps by discussing how δ PSI cut-offs affect the observed changes. For example, they might examine the dPSI ratio distributions after separating the splicing events into bins based on the CD3 dPSI.

As suggested by the reviewers, we binned AS events regulated by CD28 costimulation by CD3 dPSI levels. As shown in a new Supplemental Figure S4A, this analysis shows that CD28 signaling primarily regulates those splicing events that exhibit minimal changes with CD3 alone, as splicing events regulated by CD28 costimulation primarily have a CD3 dPSI between 0-25%.

5. The authors should provide a list of the RNA binding proteins that are upregulated under various conditions. It is not in the referenced Supplemental Table 2.

We apologize for the accidental omission and have now included a list of significantly differentially expressed RNA binding proteins regulated in various conditions (anti-CD28, anti-CD3, anti-CD3/CD28) in Supplemental Table 2.

How were the enriched motifs connected to particular proteins? It appears they could be binding sites for several possible factors, besides the ones proposed.

Enriched motifs that are connected to the proteins mentioned (SRSF1, hnRNA2B1) were done by cross referencing RBPs whose exhibit differential expression upon CD28 costimulation with RBP motifs as identified in two commonly used databases (http://cisbp-rna.ccbr.utoronto.ca, https://attract.cnic.es/index). This has now been clarified in the Methods. We acknowledge these motifs could certainly be tied to other factors and that RBPs can also be regulated in a posttranscriptional manner, so recognize that our focus on differentially expressed RBPs may miss additional regulators. However, it was not feasible for us to analyze other aspects of RBP regulation with our set of human donors and do feel that differentially expressed RBPs are highly likely to be exerting a regulatory impact. We have modified the text to clarify all of these points.

In Figure 2G, there appears to be a typo – do they mean hnRNP A1B2 or A2B1?

We have corrected the typo to hnRNPA2B1

6. In Figure 3 it would be helpful to note the starting PSI value from unstimulated cells for each gene.

We have now added the unstimulated PSI values to each gene in a revised Figure 3.

7. Figure 4F: A quantification of the changes in protein levels should be presented.

We have now added the western blot quantifications, averaged between human donors, in a new panel Figure 4G.

8. Figure 6: it is unclear why the AMO directed against the junction of exon 3 with intron 3 in CASP9 should cause skipping of all exons between 3 and 6 and thus lead to splicing between exons 2 and 7. It would be helpful to show an analysis of the patterns of splicing induced by this AMO in the region.

In regards to the mechanism of Caspase-9 splicing, the Chalfant group has previously demonstrated that exon 3 contains cis-regulatory elements that regulates the skipping of all exons 3-6 (Goehe et al., JCI 2010). While the detailed mechanisms of how the multi-skipping event is regulated remains unclear, our data showing that the AMO designed to block the junction near exon 3 also regulates the skipping of exons 3-6 is consistent with this previous work. Importantly we confirmed that there are no intermediate products seen on an RT-PCR gel after AMO transfection (gel shown in Author response image 1), while AMOs to other exons in this cassette did result in intermediate products. We have modified the text to describe this observation.

**Author response image 1. sa2fig1:** 

More generally, the effects of combining AMOs in panels B and C are described as synergistic. It appears that the effects of combining AMOs with etoposide (panel B) are additive rather than synergistic. The combined AMOs also reduce camptothecin-mediated cell death, but puzzlingly the individual AMOs seem to stimulate the camptothecin effect (in contrast with the CRISPR results from Figure 5). These results need a better description.

We agree fully with the reviewers that the effects we observe of combining AMOs is additive and we have revised the text accordingly. As for the camptothecin results, we recognize what the reviewers are noticing, but these differences are not statistically significant. We have now modified the text to clarify this.

9. In Figure 7, the exons all appear sensitive to Jnk inhibition, but their responses to other kinase inhibitors are quite distinct. The argument that they share a common regulatory pathway seems overstated. Given that the kinase inhibitor data are just an initial result and inconclusive, the authors should consider whether these data make a critical contribution to the manuscript or could be left out, pending a more thorough analysis.

Downstream of T cell receptor stimulation, there are numerous signaling pathways that are activated which are essential for T cell effector responses, such as proliferation and cytokine production. It is understood that there is cross-talk between many of these pathways, which may explain how each splicing event analyzed seems to be responsive to multiple pathways and not just a singular pathway component. In addition, alternative splicing is regulated through a combinatorial manner by multiple RNA binding proteins, such that there is potential for multiple pathways to regulate one splicing event via distinct proteins. Given all of these factors, we decided to simplify and focus on one pathway, *JNK* signaling, which is a commonality between Caspase-9, Bim, and Bax splicing events. We note that this pathway has also been shown to be activated by co-stimulation, providing a potential mechanism. We have now modified the Discussion to highlight caveats of the inhibitor data. We are willing to remove these results if the reviewers feel strongly; however, in the spirit of *eLife* in promoting scientific dialogue and future experiments, we prefer to retain this data as we feel it could be useful to direct future studies in other laboratories.